# Broad phosphorylation mediated by testis-specific serine/threonine kinases contributes to spermiogenesis and male fertility

Xuedi Zhang [1,3], Ju Peng[1,3], Menghua Wu[1,2,3], Angyang Sun[1], Xiangyu Wu[1], Jie Zheng[1], Wangfei Shi[1] & Guanjun Gao [1] ✉

Genetic studies elucidate a link between testis-specific serine/threonine kinases (TSSKs) and male infertility in mammals, but the underlying mechanisms are unclear. Here, we identify a TSSK homolog in *Drosophila*, CG14305 (termed dTSSK), whose mutation impairs the histone-to-protamine transition during spermiogenesis and causes multiple phenotypic defects in nuclear shaping, DNA condensation, and flagellar organization in spermatids. Genetic analysis demonstrates that kinase catalytic activity of dTSSK, which is functionally conserved with human TSSKs, is essential for male fertility. Phosphoproteomics identify 828 phosphopeptides/449 proteins as potential substrates of dTSSK enriched primarily in microtubule-based processes, flagellar organization and mobility, and spermatid differentiation and development, suggesting that dTSSK phosphorylates various proteins to orchestrate postmeiotic spermiogenesis. Among them, the two substrates, protamine-like protein Mst77F/Ser[9] and transition protein Mst33A/Ser[237], are biochemically validated to be phosphorylated by dTSSK in vitro, and are genetically demonstrated to be involved in spermiogenesis in vivo. Collectively, our findings demonstrate that broad phosphorylation mediated by TSSKs plays an indispensable role in spermiogenesis.

Spermatogenesis is a dynamic process that involves the proliferation and differentiation of spermatogonia, meiosis of spermatocytes, and spermiogenesis[1]. The spermiogenesis phase transforms haploid round spermatids into elongated spermatids and fully functional mature spermatozoa, which is morphologically characterized by formation of the acrosome, development of a flagellum (tail), elimination of cytoplasm, condensation of chromatin (head), an delongation of sperm cells[2]. At the chromatin level, histones on premature sperm DNA are largely transiently replaced by transition proteins and subsequently by highly basic protamines via an epigenetic remodeling process known as the histone-to-protamine transition, which is critical for packaging of haploid DNA into mature sperm heads[3]. Spermiogenesis is strictly regulated by complex machinery. Aberrant spermiogenesis negatively affects the quantity, motility, and morphology of sperm, and the ability of sperm to fertilize an egg.

Due to the inactive transcriptional machinery during spermatogenesis, sperm cell development not only requires a few testis-associated proteins but is also highly dependent on different post-translational modifications (PTMs)[4]. Only a few PTMs have been identified in sperm that are of potential importance to sperm physiology[4]. For example, protein ubiquitination has been extensively studied during spermatogenesis and regulates sperm maturation[5]. RNF8-mediated histone ubiquitination epigenetically regulates nucleosome removal during spermiogenesis by inducing H4 acetylation in trans to initiate the histone-to-protamine transition[6]. Multiple human sperm proteins identified as targets of nitrosylation are involved in spermiogenesis[7]. In addition, small ubiquitin-like modifiers

[1]School of Life Science and Technology, ShanghaiTech University, 201210 Shanghai, China. [2]School of Life Sciences, Tsinghua University, 100084 Beijing, China. [3]These authors contributed equally: Xuedi Zhang, Ju Peng, Menghua Wu. ✉e-mail: gaogj@shanghaitech.edu.cn

(SUMOs) are highly enriched in human, mouse, and fly sperm, and the SUMO modifications they mediate play crucial roles in nuclear shaping and spermatid elongation[8].

Another type of PTM, kinase-mediated phosphorylation, has emerged as a critical regulatory mechanism during spermatogenesis that participates in the entire process of germ cell development, from spermatogonia to spermatids, including subsequent sperm capacitation, motility, the acrosome reaction, and fertilization processes. Some protein kinases such as mitogen-activated protein kinases (MAPKs), POLO-like kinases (PLKs), and aurora kinase C (AURKC) are involved in multiple stages of spermatogenesis and are essential for mouse fertility[9–13]. The testis-specific serine/threonine kinase (TSSK) family is an important protein kinase family whose members are predominantly expressed in testes and function in spermiogenesis[14–19]. The TSSK family comprises six members: TSSK1, TSSK2, TSSK3, TSSK4, and TSSK6 are conserved in mouse and human, while TSSK5 is protein coding in the mouse and is a pseudogene in human[20]. Knockout mouse models for some of these genes have been created and found to affect male fertility. Tssk1/Tssk2 double KO, Tssk3 and Tssk6 single KO mice display infertility, while Tssk4 single KO mice display subfertility[21–25]. Testis-specific kinase substrate (TSKS), myelin basic protein (MBP), cAMP-responsive element-binding protein (CREB), outer dense fiber 2 (ODF2), and histones are reportedly specific substrates of TSSK in vitro[18,21,24,26–28]. However, little is known about functions of TSSK-mediated protein phosphorylation and/or phospho-modification in spermatids during spermiogenesis.

Similar to mammals, spermiogenesis in *Drosophila* involves a series of phase-separated morphological changes to form mature sperm, during which histones bound to spermatid chromatin are replaced by several highly basic MST-HMG-box proteins including the transition protein Tpl94D, Mst33A, the protamine-like protein Mst77F, and protamines (Mst35Ba and Mst35Bb)[29]. Among them, Mst77F plays a central role in spermatid nuclear shaping/condensing during spermiogenesis and male fertility[30,31]. Due to its readily identifiable stereotypical morphology, distinct stages, highly conserved functional genes with humans, and easy genetic manipulation, *Drosophila* has long been used as a model system to investigate the molecular underpinnings of PTMs in spermiogenesis and fertility[32].

Here, we show that *Drosophila* CG14305 encodes a functional homolog of human TSSKs and screen its potential substrates by performing quantitative phosphoproteomic analysis. Using a combination of genetic and biochemical analysis, we validate that CG14305 phosphorylates a few potential substrates in vitro and illustrate their biological roles in spermiogenesis in vivo. Furthermore, we find that CG14305 mutations severely impair the histone-to-protamine transition during spermiogenesis, consistent with the phenotypic defects in spermatid nuclear morphology, and demonstrate that TSSK-mediated phosphorylation plays indispensable roles in *Drosophila* spermiogenesis, shedding light on the mechanisms underlying TSSK-associated male infertility.

## Results

### *Drosophila* CG14305 deficiency results in male infertility

*Drosophila* CG14305 was predicted to be an ortholog of human TSSKs[33,34]. Both modENCODE RNA-sequencing data and developmental proteome data in FlyBase (www.flybase.org) demonstrated that CG14305 is predominately expressed in testes, similar to mammalian TSSKs[35–37]. To further investigate the degree of conservation of *Drosophila* CG14305 with human TSSKs, we first compared the amino acid sequence of CG14305 with those of all five TSSKs in humans (Fig. S1a)[38]. Multiple sequence alignments revealed that CG14305 shared 37–46% identity and 54–66% similarity with human TSSKs (Fig. S1b). Phylogenetic analysis classified CG14305 into the human TSSK1B and TSSK2 subgroup (Fig. S1c), suggesting that it might be more conserved with TSSK1B and TSSK2 subgroup. Based on the alignment results,

CG14305 possesses essential kinase characteristics, including an ATP-binding region (34−60 aa), a potential protein kinase active site (150−162 aa), and a potential activation loop phosphorylation motif (190−192 aa). Furthermore, structural prediction in the AlphaFold Protein Structure Database[39] showed that the structure of the S_TKc kinase catalytic domain of CG14305 (28−291 aa) is very similar to the structures of the counterparts of human TSSKs (Figs. S2a and S2b). Given the sequence homology and structural similarity, we refer to CG14305 as dTSSK hereinafter.

To elucidate the biological function of dTSSK, we first examined its expression and localization in different tissues. Western blot (WB) analysis of dTSSK transgenetic flies tagged with Flag-GFP and driven by its endogenous promoter shows that dTSSK was specifically expressed in testicular tissue (Figs. 1a and S3a). Imaging of whole testes revealed that dTSSK signals exclusively localized to postmeiotic spermatid bundles, with bright spots scattered over the bundles (Fig. 1b). Further dissection of testicular tissue revealed that these bright spots were mainly located on the nucleus, flagellum, and individualization complex (IC) of spermatids (Fig. S3b). To better discern the subcellular localization of dTSSK in spermatids, we investigated protein localization throughout spermiogenesis, from the round stage to the needle stage. dTSSK signals first appeared around the basal body at one side of the nucleus, as evidenced by their colocalization with PACT (a centriole-targeting domain found in *Drosophila* pericentrin-like protein, D-PLP) in PACT-mCherry transgenic flies[40], and then extended along one side of the spermatid nucleus and spread throughout the nucleus (Figs. 1c and S3c). The intensity of dTSSK signals peaked in the late canoe stage, and a low level of signals was retained until the needle stage (Figs. 1c and S3b). These results imply that dTSSK is involved in multiple spermiogenesis processes.

To further define the roles of dTSSK in spermiogenesis, the endogenous dTSSK gene was disrupted using CRISPR/Cas9 technology. Coinjection of Cas9-mRNA and dTSSK-specific gRNAs into *Drosophila* embryos, followed by germline transformation, resulted in generation of dTSSK-knockout (dTSSK$^{-/-}$) flies with a premature stop codon that completely removed the S_TKc kinase catalytic domain of dTSSK (Fig. 1d). Mutagenesis of the dTSSK gene was further confirmed by genomic PCR sequencing (Fig. 1d). WB revealed that dTSSK expression was undetectable in dTSSK$^{-/-}$ flies (Figs. 1e and S3d). Adult dTSSK$^{-/-}$ flies were morphologically similar to wild-type flies but exhibited male-specific infertility (Fig. 1f). To investigate whether dTSSK$^{-/-}$ flies could produce mature sperm, we generated transgenic flies in which the protamine Mst35Bb was tagged with GFP[41]. By tracking GFP fluorescence, we detected no mature spermatozoa in seminal vesicles in testicular tissue of dTSSK$^{-/-}$ male flies (Fig. 1g), which explained their sterility. To exclude the possibility that the infertility of dTSSK$^{-/-}$ male flies was caused by off-target events, in-trans rescue experiments were performed using dTSSK$^{-/-}$ flies that expressed transgenic wild-type dTSSK under the control of its endogenous promoter. Infertility of dTSSK$^{-/-}$ male flies was completely rescued by transgenic expression of dTSSK, indicating that it is caused by dTSSK deficiency (Fig. 1f and g).

### dTSSK depletion causes severe spermiogenesis defects

To determine why dTSSK$^{-/-}$ male flies fail to produce mature sperm, we cytologically examined cell morphological changes during spermiogenesis. The absence of dTSSK did not markedly affect nuclear morphology at the round, elongating, and early canoe stages (Fig. 2a). At the late canoe stage or before 64-cell individualization, nuclear bundles of mutant spermatids were severely curled, disorganized, and less condensed compared with tightly clustered control spermatids (Figs. 2a and S4a), indicating that dTSSK plays a critical role in sperm DNA compaction. Considering that the basal body functions as a microtubule-organizing center and the nucleating assembly of the spermatid flagellar axoneme during flagellum/cilium elongation[29,42],

 2

we then investigated whether dTSSK disruption affects embedding of the basal body on sperm nuclei. Imaging of living PACT-GFP transgenic flies revealed that the GFP signal at the basal body was massively reduced in dTSSK[-/-] male flies at the late canoe stage (Fig. 2b), indicating that the basal body anchored to sperm nuclei is partially degenerated during spermiogenesis. Consistently, flagella were obviously disorganized in spermatid bundles of dTSSK[-/-] male flies (Figs. 2c and S4b), suggesting that dTSSK influences the structural integrity of flagella. In addition, we stained testicular tissue with fluorescently labeled phalloidin to reveal actin-based ICs[43]. In contrast with the condensed and well-organized ICs in wild-type testes, we did not observe IC formation after the late canoe stage in dTSSK[-/-] testes (Figs. 2d and S4a), demonstrating that spermiogenesis was blocked during or after the late canoe stage in dTSSK[-/-] male flies. Taken together, these results reveal that dTSSK might be involved in multiple processes of spermatogenesis, including morphology of spermatid nuclei, flagellar organization, individualization progression, and sperm maturation.

## dTSSK depletion impairs the histone-to-protamine transition

The histone-to-protamine transition during spermatid chromatin remodeling is a dynamic process that occurs in a stepwise fashion,

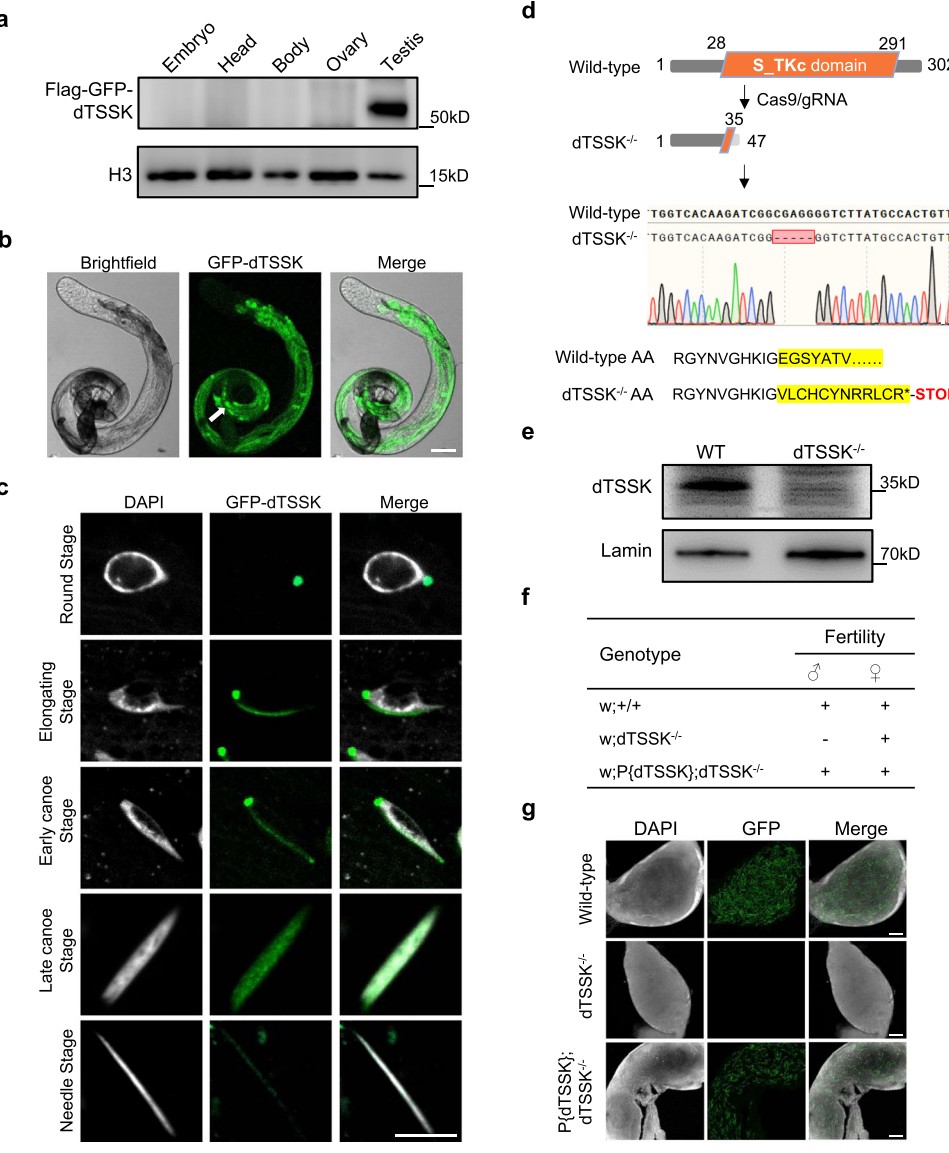

**Fig. 1 | *Drosophila* CG14305 is essential for male fertility. a** WB analysis showing the specific expression of the dTSSK protein in testicular tissue. Different tissues (including embryo, head, body, ovary, and testis) were dissected from dTSSK transgenetic flies tagged with Flag-GFP and driven by its endogenous promoter. Tissue homogenates were used for WB analysis against anti-Flag, and the observed band size of Flag-GFP-dTSSK is consistent with its predicted size. H3 was used as a loading control. **b** Live imaging showing the distribution of dTSSK protein in testes of GFP-dTSSK transgenic flies (GFP-dTSSK, green). Sperm bundles are indicated by the white arrow. Scale bar, 100 μm. **c** Live imaging showing the distribution of dTSSK protein during spermiogenesis from the round state to the needle stage (sperm DNA stained with DAPI, white; GFP-dTSSK, green). Scale bar, 5 μm. **d** CRISPR/Cas9-mediated dTSSK gene disruption in *Drosophila*. Sequencing results revealed the deletion of five bases in the coding region of the dTSSK gene, resulting in formation of a premature stop codon and complete removal of the S_TKc kinase catalytic domain. **e** WB analysis showing depletion of dTSSK protein in dTSSK[-/-] testicular extracts. The band size of dTSSK protein in wild-type flies is consistent with its predicted size of 35 kD. Lamin was used as a loading control. **f** Fertility testing of wild-type (w[1118]), dTSSK[-/-], and w;P{dTSSK};dTSSK[-/-] flies. dTSSK[-/-] flies exhibit male sterility, and replenishment of wild-type dTSSK protein restores male fertility. **g** Live imaging of seminal vesicles of the indicated genotypes stained with DAPI (white). Sperm nuclei (green) are labeled with protamine Mst35Bb-GFP. No mature sperm are detected in dTSSK[-/-] flies. Scale bar, 30 μm. Source data are provided as Source Data file.

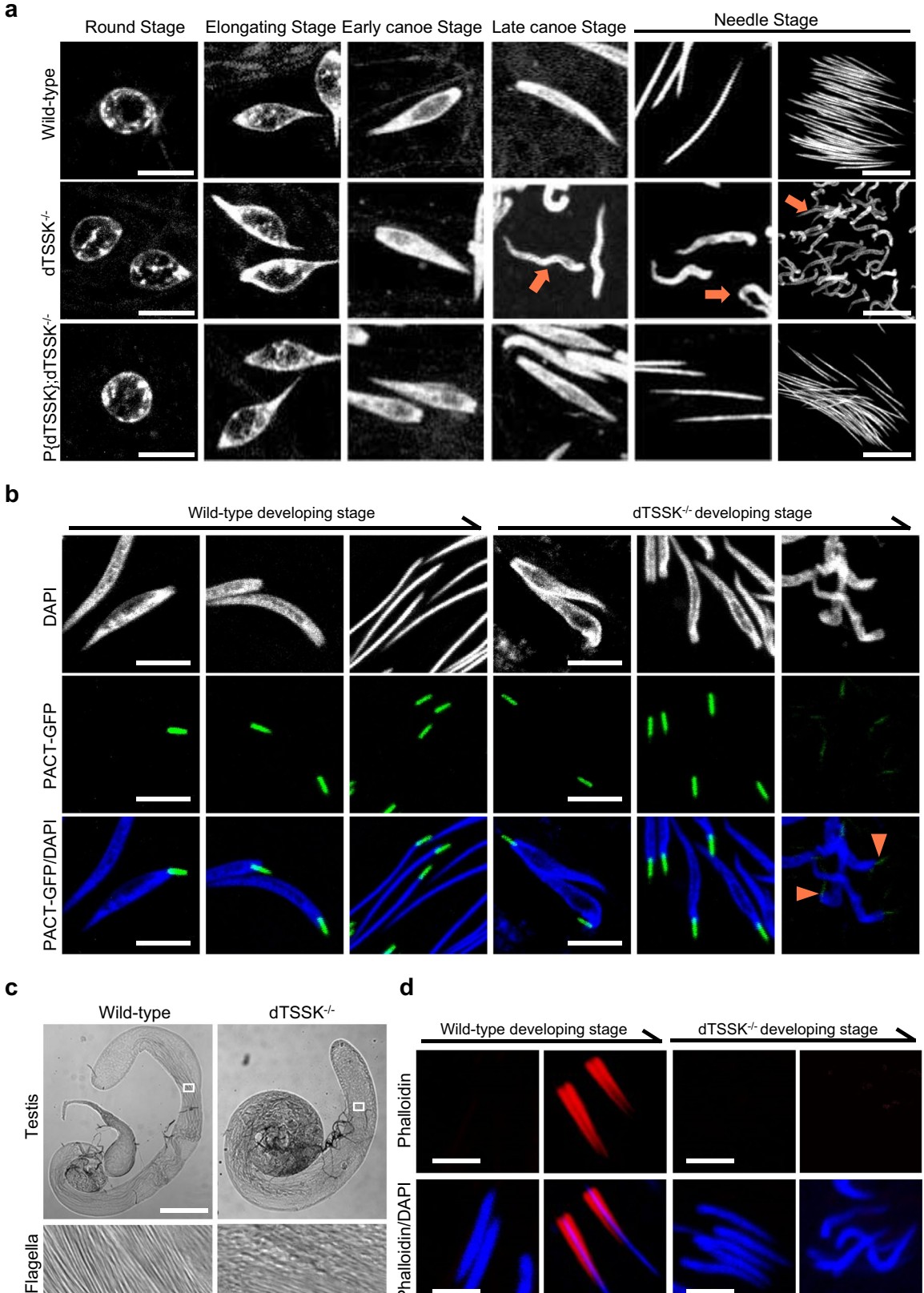

including histone expulsion, transition protein incorporation/removal, and protamine incorporation. Errors during the histone-to-protamine transition can impair sperm chromatin condensation and sperm quality, and even male sterility[44]. Since apparent chromatin decondensation in spermatids of dTSSK[-/-] flies at the late canoe stage, we further investigated whether there is any defect during histone-to-

protamine transition in these flies. We generated a series of transgenic flies, including those expressing GFP-tagged histones H2A, H2B, H3, and H4, and the transition protein Tpl94D and mCherry-tagged protamine Mst35Bb, to track the histone-to-protamine transition. Live imaging showed that H2A, H2B, H3, H4, and Tpl94D were progressively removed from sperm chromatin and replaced by Mst35Bb as

**Fig. 2 | *Drosophila* dTSSK is crucial for spermiogenesis. a** Sperm nuclear morphology at different stages of spermiogenesis in wild-type (w[1118]), dTSSK[−/−], and dTSSK-rescued dTSSK[−/−] flies. The right panel at the needle stage shows whole-mount 64-cell spermatids. Scale bar, 10 μm. Spermatid nuclei of dTSSK[−/−] flies have a curled, decondensed, and disorganized morphology (orange arrows) in/after the late canoe stage. DNA stained with DAPI, white. Scale bar, 5 μm. **b** Live imaging showing degeneration of basal body localization (orange arrowhead) in final-stage spermatids of dTSSK[−/−] male flies. PACT was used as a marker of the basal body. PACT-GFP fluorescence (green) is detected in spermatid nuclei from the early canoe stage to the final developmental stage in testes of wild-type and dTSSK[−/−] flies. Scale bar, 5 μm. **c** Cytological examination showing defects of flagellar arrangement in dTSSK[−/−] male flies. The upper panel shows the morphology of the whole testis in wild-type (w[1118]) and dTSSK[−/−] flies. The lower panel shows the arrangement of a small patch of flagella (white rectangle) in testicular tissue. **d** Phalloidin staining of spermatids of w[1118] and TSSK[−/−] flies showing the failure of IC formation after dTSSK depletion. ICs stained with phalloidin, red. Nuclei stained with DAPI, blue. Scale bar, 5 μm.

spermiogenesis progressed in the wild-type background (Figs. 3a–c, and S5a–S5c). Similarly, both H2A and H2B were removed from early-developing sperm chromatin in the dTSSK[−/−] background (Fig. S5b and S5c). However, H3, H4, and Tpl94D were retained on sperm DNA and not removed from spermatid nuclei in the dTSSK[−/−] background, without significantly affecting the incorporation of Mst35Bb (Fig. 3a–c). In addition, the histone variant H2Av was properly removed from DNA in spermatids of dTSSK[−/−] male flies (Fig. S6a), whereas H3.3 was retained in nuclei at the final sperm stage in these flies (Fig. S6b). Collectively, these results demonstrate that dTSSK deficiency causes significant defects in complete histone eviction and transition protein removal during histone-to-protamine transition.

### Human TSSKs rescue the defective nuclear morphology of dTSSK[−/−] spermatids

To further investigate the functional conservation of TSSKs during spermiogenesis across various species and determine whether human TSSKs can rescue the defective phenotype of dTSSK[−/−] flies during spermiogenesis, we generated transgenic flies in which human TSSK1B, TSSK2, TSSK3, TSSK4 and TSSK6 were respectively expressed under the control of the endogenous dTSSK promoter. After crossing with dTSSK[−/−] flies, the five types of human TSSK-rescued flies remained infertile and lacked mature sperm in seminal vesicles (Fig. S7a). Next, we examined the sperm nuclear morphology in these flies. Our findings revealed that the sperm nuclei of TSSK2-, TSSK3-, and TSSK4-rescued flies remained abnormal, while TSSK1B- and TSSK6-rescued flies exhibited sperm nuclei morphology that was similar to that of wild-type flies (Fig. S7b). Cytological examination revealed that H2A, H2B, H3, H4, and Tpl94D were normally removed from sperm chromatin in TSSK1B-rescued flies, similar to wild-type flies (Fig. S8a). In addition, flagellar alignments were partially restored in TSSK1B-rescued flies (Fig. S8b). However, we did not detect normal IC formation in spermatids of TSSK1B-rescued male flies (Fig. S8c). This probably explains the lack of mature sperm in seminal vesicles of TSSK1B-rescued flies because IC formation is an essential membrane remodeling process for 64-cell spermatids to form mature spermatozoa in *Drosophila*[45]. Taken together, these results suggest that the biological function of TSSKs is conserved from flies to humans, although dTSSK also contributes to spermatid individualization.

### The N- and C-terminal domains of dTSSK are both essential for male fertility

Human TSSKs did not rescue male infertility of dTSSK[−/−] flies; therefore, we wondered whether the N-terminal (1–26 aa) and C-terminal (292–302 aa) domains of dTSSK, which are poorly conserved with human TSSKs (Figs. S1a and S2a), are critical for spermiogenesis in *Drosophila*. We generated transgenic flies expressing dTSSK[ΔN] (deleted N-terminal 26 aa) and dTSSK[ΔC] (deleted C-terminal 11 aa). These two types of transgenic flies in the dTSSK[−/−] background both exhibited male sterility. Cytological examination revealed no mature sperm in seminal vesicles of dTSSK[ΔN]- and dTSSK[ΔC]- rescued flies (Fig. S9a). Live imaging of GFP-tagged dTSSK[ΔN] transgenic flies revealed that dTSSK[ΔN] exclusively localized to postmeiotic spermatid bundles, similar to wild-type dTSSK, but with weak signals in sperm nuclei (Fig. S9b and S9c), suggesting that the N-terminal domain of dTSSK affects protein

localization to sperm nuclei. Consistently, although spermatids of dTSSK[ΔN]-rescued flies developed a needle-shaped morphology, some sperm displayed abnormal DNA compaction (Fig. S9d), indicating that spermiogenesis can progress to a later stage upon expression of dTSSK[ΔN], although ICs did not form (Fig. S9e). Compared with dTSSK[ΔN], the phenotype of dTSSK[ΔC]-rescued flies was better, with obvious GFP signals observed in sperm nuclei (Fig. S9c) and a higher proportion of needle-shaped spermatids with a normal morphology (Fig. S9d), although ICs still did not form (Fig. S9e). Collectively, these findings suggest that N- and C-terminal domains of dTSSK are both essential for sperm mature and male fertility.

### The kinase catalytic activity of dTSSK plays a critical role in spermiogenesis

To determine whether dTSSK regulates spermiogenesis through its kinase activity in vivo, we generated a GFP-dTSSK[K60M] transgenic fly expressing a kinase-dead transcript in which the conserved Lys60 in the ATP-binding site of the S_TKc kinase domain was substituted by methionine (Fig. S2b)[28,46,47]. Live imaging revealed that dTSSK[K60M] signals exclusively localized to postmeiotic spermatid bundles, but there were no strong puncta in spermatid nuclei as in wild-type flies (Fig. S10a and S10b), suggesting that the kinase activity of dTSSK plays an essential role in its localization to sperm nuclei. To precisely define the function of dTSSK[K60M], we generated native dTSSK[K60M] transgenic flies without any tags. After crossing with dTSSK[−/−] flies, we found that dTSSK[K60M] flies in the dTSSK[−/−] background exhibited male sterility and lacked mature sperm in seminal vesicles (Fig. S10c). Moreover, curled and decondensed nuclear bundles were observed during the final nuclear morphogenesis of dTSSK[K60M] mutants at the late canoe stage (or before the IC stage), similar to dTSSK[−/−] male flies (Fig. 4a). In addition, live imaging revealed that the dTSSK[K60M] mutation caused obvious basal body degeneration in the final stage of spermiogenesis (Fig. 4b). Flagella of dTSSK[K60M] flies displayed a slightly irregular arrangement (Fig. S10d). Similar to dTSSK[−/−] flies, H3, H4, and Tpl94D were not removed from sperm DNA at the final stage in dTSSK[K60M] flies (Figs. 4c and S11), suggesting that the kinase activity of dTSSK is necessary for the histone-to-protamine transition. Next, we generated two dTSSK[T191A] transgenic flies with and without GFP marker to determine whether another kinase dead site dTSSK-Thr191 in the conserved potential activation loop phosphorylation motif engaged in spermiogenesis regulation[47]. Similar to dTSSK[−/−] and dTSSK[K60M]-rescued flies, dTSSK[T191A] failed to localize to spermatid nuclei (Fig. S10a and S10b), and dTSSK[T191A] flies exhibited severe phenotypic defects during spermiogenesis (Figs. 4a–c, S10c, S10d and S11), suggesting that the activation loop phosphorylation motif is essential for the function of dTSSK. Taken together, these results suggest that the kinase catalytic activity of dTSSK plays an essential role in normal spermiogenesis and male fertility.

### Phosphoproteomic screening identifies physiological potential substrates of dTSSK

The kinase activity of dTSSK is essential for spermiogenesis; therefore, we next identified its potential phosphorylated substrates by investigating differentially phosphorylated sites in the testes of wild-type (w[1118]) and dTSSK[−/−] male flies using proteomic and quantitative

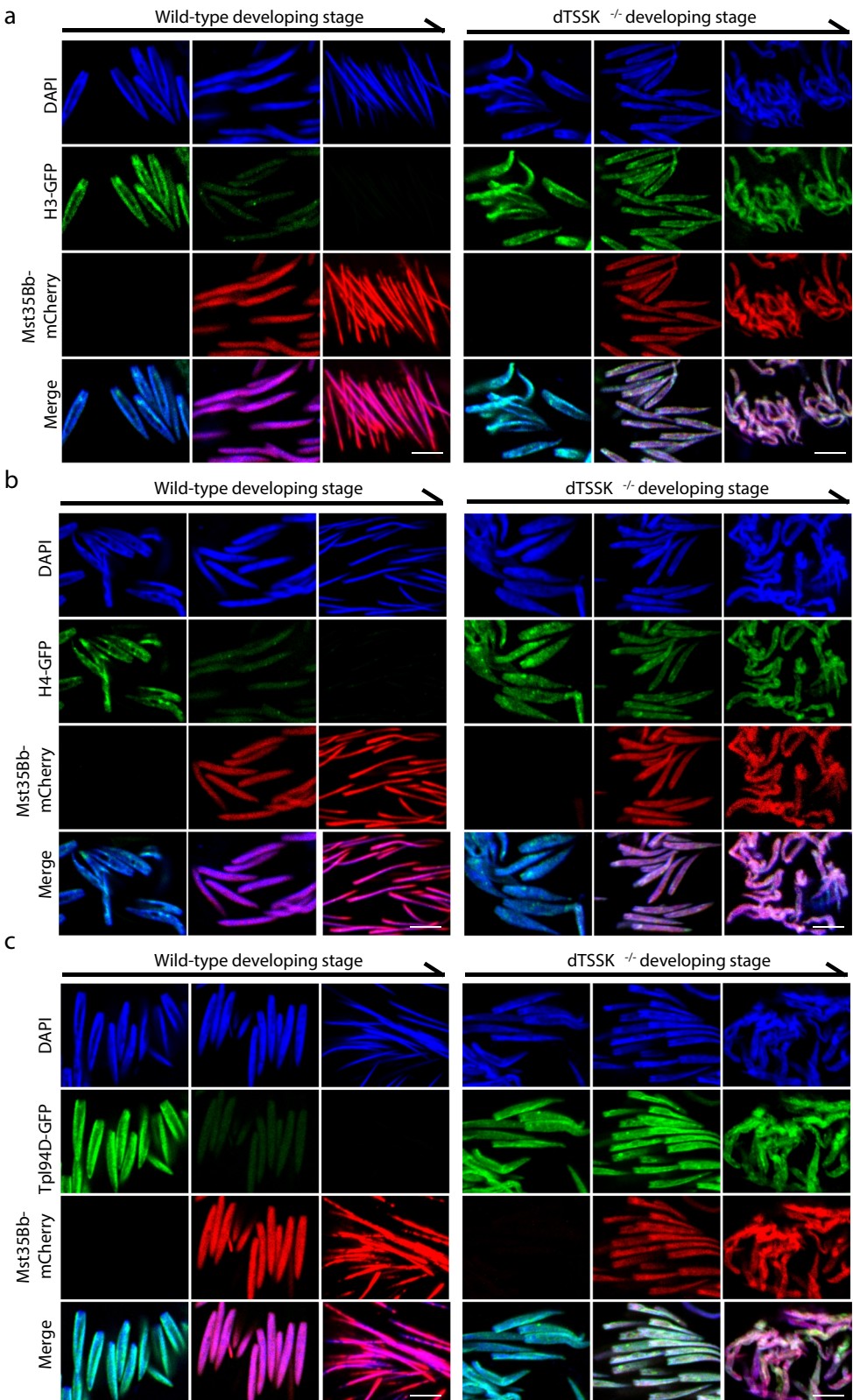

**Fig. 3 | dTSSK depletion impairs the histone-to-protamine transition. a–c** Live imaging showing defective removal of histone H3 (**a**), histone H4 (**b**), and the transition protein Tpl94D (**c**) from sperm DNA in dTSSK[−/−] flies. w[1118] flies were used as a wild-type control. H3, H4, and Tpl94D were labeled with GFP, and Mst35Bb was labeled with mCherry. Sperm images were mainly collected according to the developmental stages of spermatozoa. DNA stained with DAPI, blue; H3-GFP, H4-GFP, and Tpl94D-GFP, green; Mst35Bb-mCherry, red. Scale bar, 5 μm.

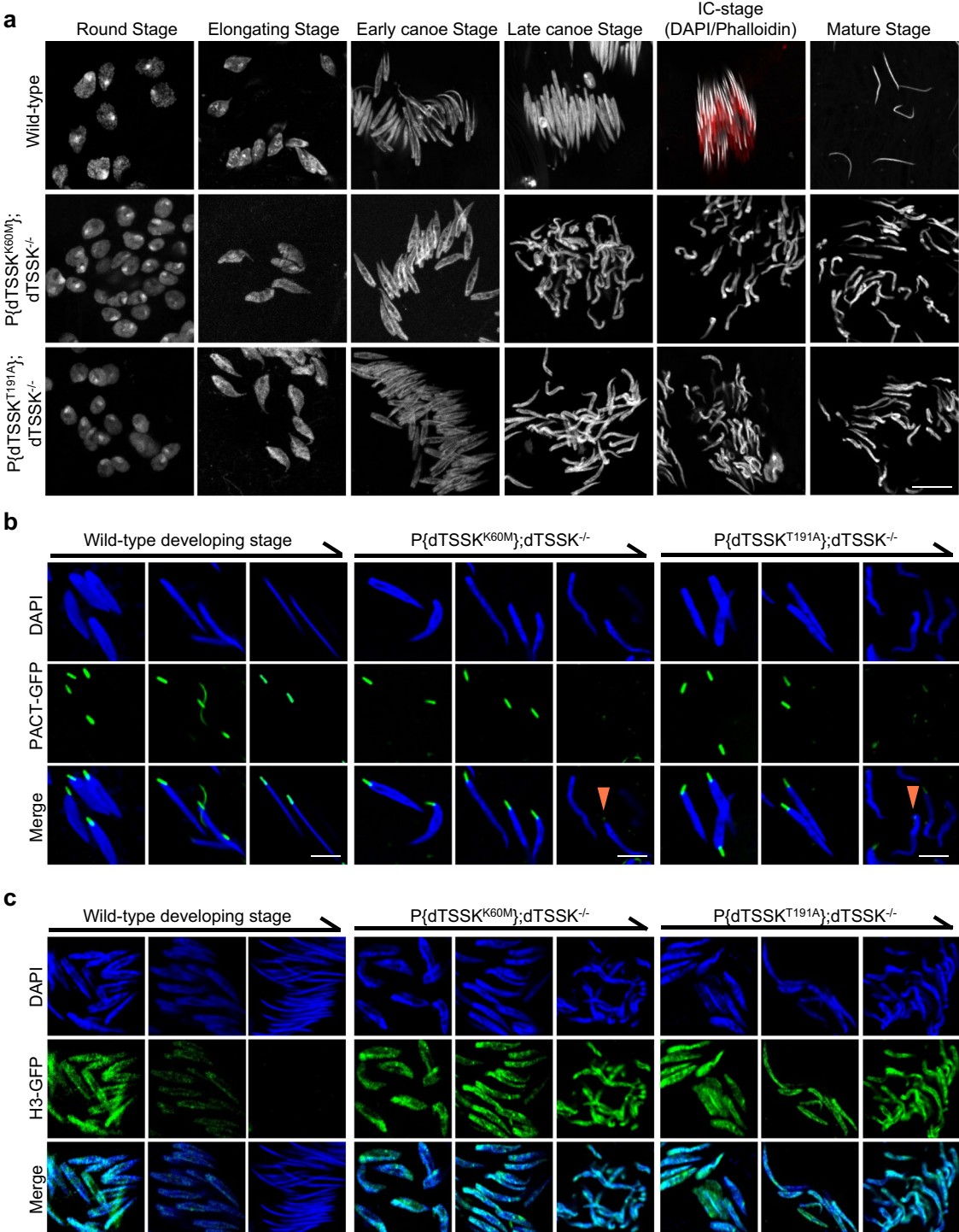

**Fig. 4 | Kinase catalytic activity of dTSSK is essential for spermiogenesis.**
**a** Sperm nuclear morphology at different stages of spermiogenesis in wild-type
(w[1118]), dTSSK[K60M], and dTSSK[T191A] flies. Lys60 or Thr191 mutation of dTSSK leads to
sperm DNA decondensation and failure of IC formation. Sperm nuclei stained with
DAPI, white. ICs stained with phalloidin, red. Scale bar, 10 μm. **b** Live imaging
showing the degeneration of basal body localization (orange arrowhead) in final-

stage spermatids of dTSSK[K60M] and dTSSK[T191A] flies. PACT was used as a marker of
the basal body. PACT-GFP fluorescence (green) is detected in spermatid nuclei
from the early canoe stage to the final developmental stage. w[1118] flies were used as
a control. Scale bar, 5 μm. **c** Live imaging showing the defective removal of histone
H3 from spermatid nuclei in dTSSK[K60M] and dTSSK[T191A] flies. H3-GFP, green; DNA
stained with DAPI, blue. Scale bar, 5 μm.

phosphoproteomic analysis. For phosphoproteomics, three biological
replicates of dTSSK[−/−] or wild-type testes were obtained, cysteines were
reduced and alkylated, and protein extracts were trypsinized. The six
samples were isotopically labeled using tandem mass tags (TMTs)
which allows quantitative, multiplexed, parallel analysis of tryptic
peptides in all six samples with high accuracy[48]. Phosphopeptides were

enriched and analyzed by LC-MS/MS on a high-resolution mass spec-
trometer (Fig. 5a). Reproducibility between replicates was high (Fig.
S12a). Integrative phosphoproteomic analyses identified 8355 unique
phosphosites in all replicates at a false-discovery rate (FDR) of <1% (Fig.
S12b). We identified 916 significant differential phosphopeptides in
dTSSK[−/−] testes compared with wild-type testes (fold-change > 1.5-fold,

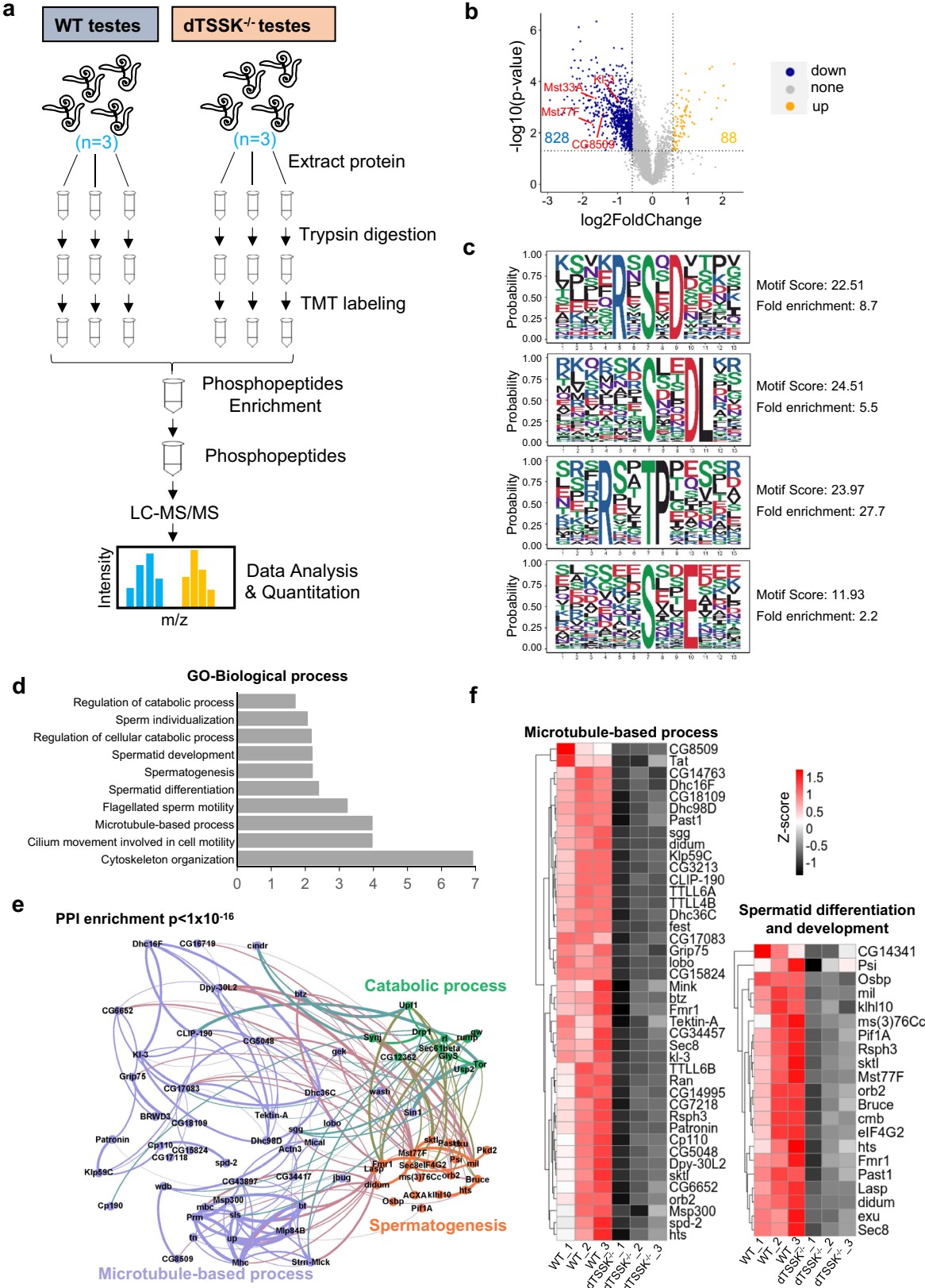

*P* < 0.05), of which 828 exhibited decreased phosphorylation, and 88 exhibited increased phosphorylation (Fig. 5b and Supplementary Data 1). The former were assigned to 485 proteins, which are potential dTSSK substrates. For proteomic profiling, we acquired 4226 proteins totally and found 563 proteins were differentially expressed (309 up-regulated and 254 downregulated, fold-change > 1.5-fold, *P* < 0.05) in

dTSSK mutant testes compared to wild-type testes (Supplementary Data 2). Motif analysis of the identified potential substrates suggested that dTSSK is an S/T kinase (Fig. 5c), similar to mammalian TSSKs[24]. To gain more insight into the functions associated with this disruption of basal phosphorylation, we performed Gene Ontology (GO) analysis. For the phosphopeptides that exhibited decreased phosphorylation in

**Fig. 5 | Phosphoproteomic screening identifies physiological substrates of dTSSK. a** Phosphoproteomics workflow. Wild-type (w[1118]) and dTSSK[−/−] testes were lysed, and protein extracts were digested with trypsin. After peptides were isotopically labeled with TMT and combined, phosphopeptides were enriched, fractionated by reversed-phase chromatography, and analyzed by quantitative mass spectrometry on an Orbitrap mass spectrometer. Three biological duplicates were performed. **b** Volcano plot showing potential dTSSK substrates. The horizontal cut-off line represents a *P* value of 0.05, and the vertical cut-off lines represent a log2 ratio of 0.58 (-1.5-fold) above which peptides were considered to differ significantly in abundance between wild-type and dTSSK[−/−] testes. Statistical analyses were performed by two-sided Student's *t* test. **c** Motifs that are enriched among phosphosites that are significantly less phosphorylated in dTSSK[−/−] testes than in wild-type testes are indicated. Favored amino acids at corresponding positions are shown in larger font, while disfavored amino acids are shown in smaller font. **d** GO term enrichment of phosphopeptides that are less abundant in dTSSK[−/−] testes than in wild-type testes. **e** Interaction network of proteins harboring phosphosites that are downregulated in dTSSK[−/−] testes compared with wild-type testes. These are part of protein complexes involved in a wide range of biological functions. Interactions were based on the STRING database and visualized by Gephi. The edge weight is proportional to the combined interaction score. Proteins (node) and the related interaction (edge) belonging to selected pathways were highlighted as indicated in the figures. **f** Heatmap of proteins belonging to the indicated pathway. The color corresponds to the normalized enrichment score (*Z*-score). Red denotes high phosphorylated abundance, and black denotes low phosphorylated abundance.

dTSSK[−/−] testes, the most enriched pathways shared by these phosphoproteins were related to various spermatogenetic processes including microtubule-based process, cilium/flagellum-dependent sperm motility, spermatid differentiation/development, sperm individualization, and catabolic process (Fig. 5d and e). Proteins involved in microtubule-based process and spermatid differentiation/development whose phosphorylation was downregulated in dTSSK[−/−] testes are shown and quantified in Fig. 5f. Interestingly, mutants of many of these proteins exhibit male sterility (Table S1), including Mst77F[49], cmb (Human homolog: PCM1)[50], kl-3 (Human homolog: DNAH)[51], ms(3) 76 Cc[52], and didum (Human homolog: Myosin V)[53]. Taken together, these data suggest that dTSSK-mediated phosphorylation participates in many spermiogenesis processes, consistent with its subcellular localization and the morphological defects of spermatids observed in dTSSK[−/−] flies.

## dTSSK potentially regulates the phosphorylation of Mst77F-Ser9

Mst77F not only is a chromatin component of mature sperm but also associates with microtubules involved in the nuclear-shaping process during spermiogenesis[49]. Phosphoproteomics identified seven phosphosites in four phosphopeptides of Mst77F (Fig. 6a). Among these phosphosites, phosphorylation at Ser9 of Mst77F was most significantly decreased in dTSSK-/- flies (Fig. 6b and c), while proteomic profiling did not reveal the decreased protein expression of Mst77F (Supplementary Data 2). To evaluate whether dTSSK regulates the phosphorylation of Mst77F-Ser9 in flies, antibodies (Mst77F-pSer[9]) were raised in rabbits against the synthesized phosphopeptide corresponding to the sequence around Mst77F-Ser9. Dot blot analysis revealed that affinity-purified antibodies from the bleeds recognized the phosphopeptide immunogen but hardly recognized the nonphosphopeptide equivalent (Fig. S13a). We next used the Mst77F-pSer[9]-recognizing antibodies to examine whether dTSSK regulates Mst77F phosphorylation in vivo. WB analysis revealed that endogenous Mst77F was strongly phosphorylated in whole extracts of wild-type testes, but its phosphorylation was undetectable in dTSSK[−/−] testes (Fig. 6d). We next generated transgenic flies that expressed Flag-GFP-Mst77F and demonstrated that Mst77F-Ser9 was only phosphorylated in the wild-type background, and that its phosphorylation was undetectable in the dTSSK[−/−] background (Fig. 6e). In addition, phosphorylation of Mst77F-Ser9 was abolished when protein phosphatase (Ppase) was added to the testicular lysate (Fig. 6f), suggesting that dTSSK potentially regulates the phosphorylation of Mst77F-Ser9 in vivo. Accordingly, immunofluorescence (IF) revealed that Mst77F-Ser9 was phosphorylated in wild-type flies, and its phosphorylation was undetectable in dTSSK[−/−] flies (Fig. 6g), further indicating that dTSSK regulates Mst77F phosphorylation in vivo.

To illustrate that the S_TKc kinase activity of dTSSK is responsible for phosphorylation of Mst77F-Ser9, we investigated whether Mst77F-Ser9 is phosphorylated in two types of flies with dTSSK kinase dead mutations (dTSSK[K60M] and dTSSK[T191A]). WB analysis showed that phosphorylation of Mst77F-Ser9 was abolished in both mutants (Fig. 6h), further indicating that Lys60 and Thr191 play essential roles in the kinase activity of dTSSK. As expected, Mst77F-Ser9 was phosphorylated in dTSSK[−/−] flies expressing dTSSK[ΔN] or dTSSK[ΔC] (Fig. 6i) and in dTSSK[−/−] flies with complementation of human TSSK1B or TSSK6, the kinase domains of which are conserved with that of dTSSK (Fig. 6j).

To further investigate that dTSSK can regulate the phosphorylation of Mst77F, Flag-Mst77F was coexpressed with dTSSK-V5 in cultured *Drosophila* S2 cells. WB analysis showed that overexpression of dTSSK-V5 markedly increased the phosphorylation level of Mst77F (Fig. 6k). In addition, we enriched dTSSK from testes of dTSSK-Flag transgenic flies and performed dot blot experiments to examine whether dTSSK could phosphorylate the Mst77F-Ser9 peptide in vitro. The Mst77F-Ser9 peptide was phosphorylated in the presence of dTSSK (Figs. 6l and S13b), whereas the mutated peptide Mst77F-S9A (serine substituted by alanine) was not. In controls without dTSSK, neither the Mst77F-Ser9 nor the Mst77F-S9A peptide was phosphorylated (Figs. 6l and S13b). Taken together, these results reveal that dTSSK might phosphorylate Mst77F-Ser9 in vitro.

## Phosphorylation of Mst77F by dTSSK contributes to spermiogenesis

To illustrate the involvement of Mst77F phosphorylation in spermiogenesis, a Mst77F mutant fly was generated by CRISPR/Cas9 technology (Fig. 7a). Mst77F-specific gRNA was designed to induce a 1 bp insertion and 6 bp deletion at the targeted site, resulting in generation of flies harboring mutated Mst77F (Mst77F[−/−]) with a premature stop codon. Mst77F gene disruption was further confirmed by genomic PCR sequencing. As predicted, we detected no fluorescent signal of Mst77F-pSer[9] on sperm of Mst77F[−/−] flies (Fig. S14a). Mst77F[−/−] flies exhibited spermatids with decondensed chromatin during the final stage of sperm development and male sterility (Figs. 7b, c, and S14b), consistent with previous studies[31]. Next, we generated transgenic flies expressing wild-type Mst77F and Mst77F[S9A]. Upon transgenic supplementation of Mst77F[−/−] flies with wild-type Mst77F, male infertility was rescued, and obvious Mst77F-pSer[9] signals were detected in spermatid nuclei, whereas no Mst77F-pSer[9] signals were detected in Mst77F[S9A] flies, which had reduced fertility (Figs. 7c, S14a, and S14b). Cytological examination revealed that Mst77F[S9A] expression resulted in abnormal nuclear condensation in some spermatozoa (Fig. 7b).

To further elucidate the roles of Mst77F phosphorylation in sperm DNA compaction, we generated two types of transgenic flies with a GFP tag fused to Mst77F at the N-terminus (GFP-Mst77F) and C-terminus (Mst77F-GFP). GFP signals in GFP-Mst77F flies appeared at the early canoe stage and gradually disappeared from the late canoe stage to maturity (Fig. 7d), whereas strong GFP signals were continuously maintained until sperm maturation in Mst77F-GFP flies (Fig. 7e). This supports the hypothesis that pre-Mst77F, a precursor protein present in sperm chromatin during the canoe stage, undergoes N-terminal proteolytic processing from the late canoe stage to maturity, forming mature Mst77F responsible for mature sperm chromatin hypercompaction[31]. Indeed, IF analysis with an Mst77F-pSer[9] antibody revealed that phosphorylated Mst77F signals gradually detached from

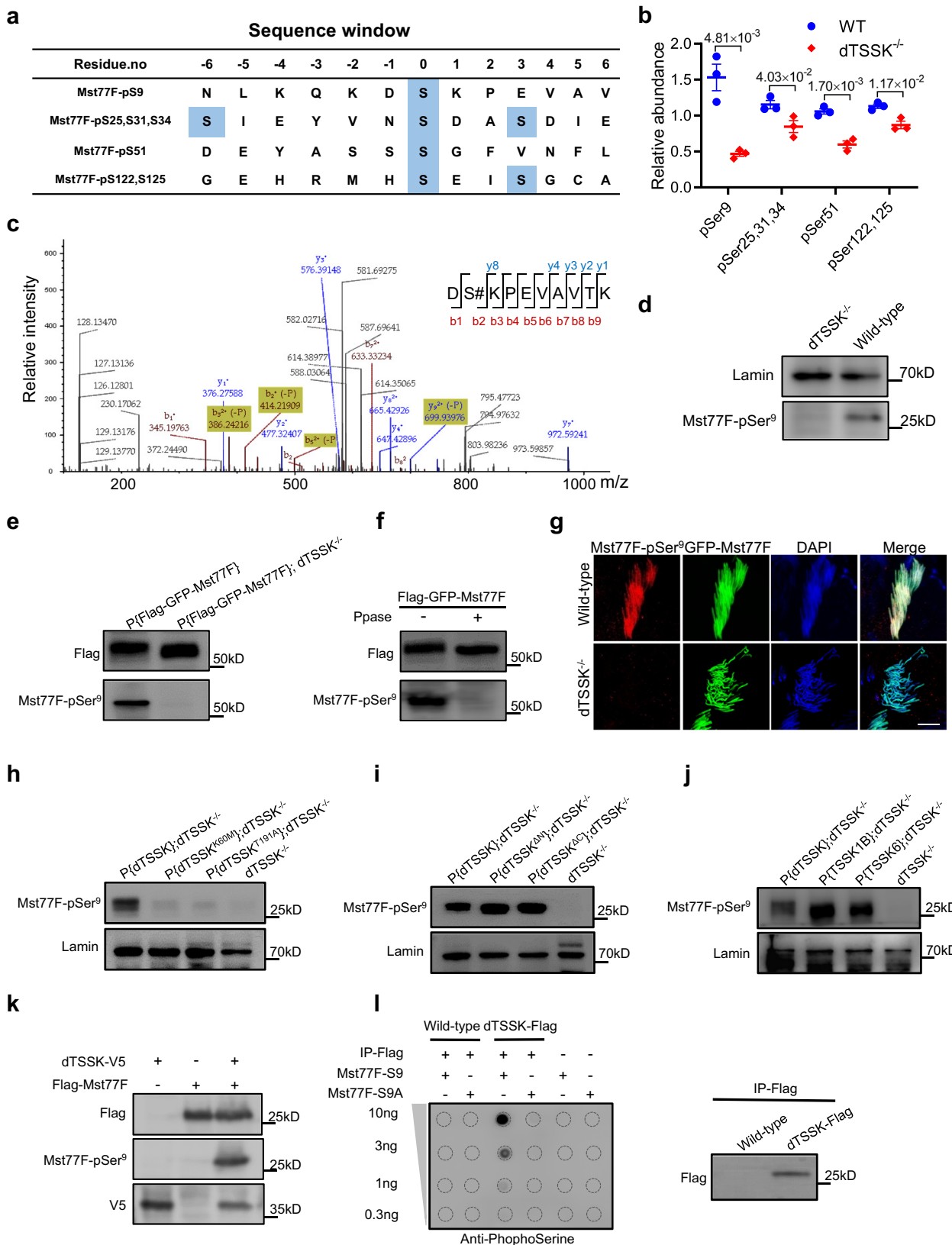

sperm DNA to form discrete spots after the late canoe stage and were completely absent in needle-shaped mature sperm (Fig. 7f). We then investigated whether phosphorylation of Mst77F-Ser9 is involved in N-terminal proteolysis of Mst77F during sperm maturation. Live imaging of GFP-Mst77F transgenic flies in the dTSSK[−/−] background (Mst77F is not phosphorylated) revealed that GFP signals persisted

until the final stages of sperm development (Figs. 7d and S14c). Moreover, we observed residual fluorescent dots on mature sperm DNA in GFP-Mst77F[S9A] transgenic flies, suggesting that Ser9 mutations affect N-terminal proteolysis of Mst77F at sperm maturation. N-terminal processing of Mst77F plays an important role in functional organization of mature sperm DNA;[31] therefore, we speculated that

**Fig. 6 | Validation of Mst77F as a substrate of dTSSK in vitro and in vivo. a** Four phosphopeptides of Mst77F protein identified by phosphoproteomic analysis. The identified phosphorylated serine residues are marked in blue. **b** Plots showing differences in the relative abundance of these four phosphopeptides of Mst77F between wild-type ($w^{1118}$) and dTSSK$^{-/-}$ flies ($n = 3$ per group). Data are mean ± SEM. Statistical analyses were performed by two-sided Student's $t$ test. **c** Detection of specific y (blue) and b (red) fragment ions allowed identification of the peptide sequence DSKPEVAVTK and assignment of the phosphorylation site to Mst77F-Ser9. Specifically, the presence of the y(8) fragment ion confirms the phosphorylation site at Ser9 of Mst77F. **d** WB analysis of testicular extracts of adult wild-type ($w^{1118}$) and dTSSK$^{-/-}$ flies using an anti-Mst77F-pSer$^9$ antibody to detect Mst77F phosphorylation. Lamin was used as a loading control. **e** WB analysis of testicular extracts expressing Flag-GFP-Mst77F protein in the wild-type ($w^{1118}$) and dTSSK$^{-/-}$ background using anti-Flag and anti-Mst77F-pSer$^9$ antibodies. The band size of the phosphorylated Flag-GFP-Mst77F fusion protein detected in the wild-type

background is consistent with its predicted molecular weight of 50 kD. **f** WB analysis showing there is no detectable phosphorylation of the Flag-GFP-Mst77F fusion protein in the wild-type background after Ppase treatment. **g** IF showing phosphorylated Mst77F-Ser9 (red) in 64-cell spermatid nuclei in wild-type flies and no detectable phosphorylation in dTSSK$^{-/-}$ flies. Mst77F was marked with GFP. Nuclei were stained with DAPI. Scale bar, 10 μm. **h–j** WB analysis of testicular extracts of indicated flies. An anti-Mst77F-pSer$^9$ antibody was used to detect Mst77F phosphorylation. Lamin was used as a loading control. **k** WB analysis of extracts of the indicated transfected *Drosophila* S2 cells. Mst77F phosphorylation is only detected in cells cotransfected with dTSSK and Mst77F. **l** dTSSK-mediated Mst77F-Ser9 phosphorylation detected by an in vitro kinase assay followed by dot blotting. Peptides corresponding to Mst77F-Ser9 are phosphorylated by purified dTSSK-Flag and detected using a commercial antiphosphoserine antibody (left panel). Detection of Flag fusion proteins shows the successful purification of dTSSK-Flag (right panel). Source data are provided as Source Data file.

phosphorylation of Mst77F-Ser9 might be involved in mature sperm DNA compaction. Collectively, these results suggest that phosphorylation of Mst77F by dTSSK is involved in spermiogenesis.

## Mst33A is a potential substrate of dTSSK involved in spermiogenesis

Phosphoproteomic analysis also revealed that Mst33A-Ser237 was less phosphorylated in the absence of dTSSK (Fig. S15a), while proteomic profiling did not reveal the decreased protein expression of Mst33A (Supplementary Data 2). We first confirmed that dTSSK could phosphorylate the peptide corresponding to Mst33A-Ser237 using a commercial antibody that recognizes Ser-specific phosphorylation by performing an in vitro kinase assay (Fig. S15b). Live imaging of GFP-Mst33A transgenic flies revealed that Mst33A predominantly localized to chromatin in canoe stage spermatozoa but was absent from needle-stage spermatozoa (Fig. S15c), suggesting that Mst33A is a sperm chromatin-associated transition protein. Disruption of Mst33A using CRISPR/Cas9 technology reduced fertility of male flies (Figs. S15d and S15e). Cytological observation showed that some spermatozoa had an abnormal nuclear morphology in Mst33A$^{-/-}$ flies (Fig. S15f). To investigate whether phosphorylation of Mst33A-Ser237 by dTSSK is involved in spermiogenesis, we generated three types of transgenic flies that expressed Mst33A, Mst33A$^{S237A}$, and GFP-tagged Mst33A$^{S237A}$ under the control of the endogenous Mst33A promoter. Live imaging showed that the Mst33A$^{S237A}$ mutation did not markedly affect the localization of Mst33A to sperm DNA (Fig. S15g). However, DNA compaction was deficient in some sperm at the needle stage in Mst33A$^{S237A}$ flies (Fig. S15f), which may correspond to a slight decrease in fertility (Fig. S15e). Taken together, these findings suggest that phosphorylation of Mst33A-Ser237 by dTSSK is involved in sperm DNA compaction.

## Autophosphorylation of dTSSK-Ser18 is essential for sperm individualization

Comparative phosphoproteomic analysis revealed significant differences in the levels of phosphorylation at Ser17 and Ser18 of dTSSK between wild-type and dTSSK$^{-/-}$ flies (Fig. 8a). To evaluate whether Ser17 and Ser18 are phosphorylated in vivo, we generated transgenic flies expressing dTSSK-Flag, dTSSK$^{S17A}$-Flag, dTSSK$^{S18A}$-Flag, and dTSSK$^{S17&S18A}$-Flag. WB analysis showed that the expression levels of dTSSK$^{S17A}$, dTSSK$^{S18A}$, and dTSSK$^{S17&S18A}$ were remarkably lower than that of wild-type dTSSK, implying that Ser17 or Ser18 mutations affect the stability of dTSSK in testes (Figs. 8b and S16a). Further analysis of whole testicular extracts by Phos-tag SDS-PAGE followed by WB analysis revealed a shifted band in extracts of dTSSK-Flag and dTSSK$^{S17A}$-Flag male flies, and only a lower band in extracts of dTSSK$^{S18A}$-Flag and dTSSK$^{S17&S18A}$-Flag flies (Fig. 8c). dTSSK appeared mainly as a phospho-modified band in dTSSK$^{S17A}$ and wild-type flies, but only as a nonphospho-modified band in dTSSK$^{S18A}$ and dTSSK$^{S17&S18A}$ flies (Figs. 8c and S16b). Supplementation of testicular extracts with Ppase

abolished the phosphorylation signals of dTSSK (Figs. 8c and S16b), further confirming that dTSSK is autophosphorylated at Ser18 in vivo.

To investigate the involvement of dTSSK-Ser18 phosphorylation in spermiogenesis, we generated transgenic flies expressing dTSSK$^{S17A}$, dTSSK$^{S18A}$, and dTSSK$^{S17A&S18A}$ without any tag. Fertility testing showed that dTSSK$^{S17A}$ male flies exhibited a slight decline in fertility, while dTSSK$^{S18A}$ male flies exhibited very poor fertility, and dTSSK$^{S17A&S18A}$ male flies were completely sterile (Fig. 8d), suggesting that Ser18 of dTSSK plays a critical role in male fertility. Next, by tracking Mst35Bb-GFP fluorescence, we observed few or no sperm in seminal vesicles of dTSSK$^{S18A}$ flies and a complete absence of sperm in dTSSK$^{S17A&S18A}$ flies (Fig. 8e), consistent with the results of fertility testing. Cytological examination of spermiogenesis revealed grossly normal sperm DNA condensation in dTSSK$^{S17A}$, dTSSK$^{S18A}$, and dTSSK$^{S17A&S18A}$ flies (Fig. 8f). Moreover, dTSSK$^{S17A&S18A}$ flies exhibited normal basal body subcellular localization similar to wild-type flies (Fig. 8g). In addition, dTSSK$^{S17A}$, dTSSK$^{S18A}$, and dTSSK$^{S17A&S18A}$ flies displayed normally organized flagellar structures (Fig. 8h). However, F-actin staining with phalloidin revealed that IC formation at the individualization stage was almost abolished in dTSSK$^{S18A}$ and dTSSK$^{S17A&S18A}$ flies (Fig. 8i), which might explain the inability of these flies to produce mature sperm. Furthermore, WB analysis revealed that the kinase catalytic activity of dTSSK was independent on Ser17 and Ser18 (Fig. 8j), consistent with the results obtained upon N-terminal deletion of dTSSK (Fig. 6i). Taken together, these results suggest that autophosphorylation of dTSSK-Ser18 contributes to spermiogenesis by facilitating the formation of ICs.

## Discussion

In this study, we identified a TSSK homolog from *Drosophila melanogaster*. Subsequent experiments found that dTSSK mutations caused male infertility, with a variety of morphological defects during sperm cell development (including those in nuclear shaping, chromatin condensation, sperm individualization, basal body localization, and sperm flagellum organization), and severely impaired sperm DNA compaction by interrupting the histone-to-protamine transition during spermiogenesis. Moreover, we confirmed the functional conservation of TSSKs from flies to humans and validated the importance and necessity of key amino acids in the kinase catalytic domain of TSSK in spermiogenesis. More importantly, state-of-the-art phosphoproteomic screening identified multiple potential substrates of dTSSK including Mst77F and Mst33A, which were demonstrated to play important roles in sperm chromatin condensation. Thus, our work demonstrates that extensive phosphorylation mediated by dTSSK might be essential for male fertility, further advancing understanding of the molecular mechanism underlying phosphorylation-based regulation by TSSKs.

### TSSKs in *Drosophila* and mammals

Transcriptome and IF analyses of mouse germ cells revealed that different TSSK family members (Tssk1, Tssk2, Tssk3, Tssk4 and Tssk6) are

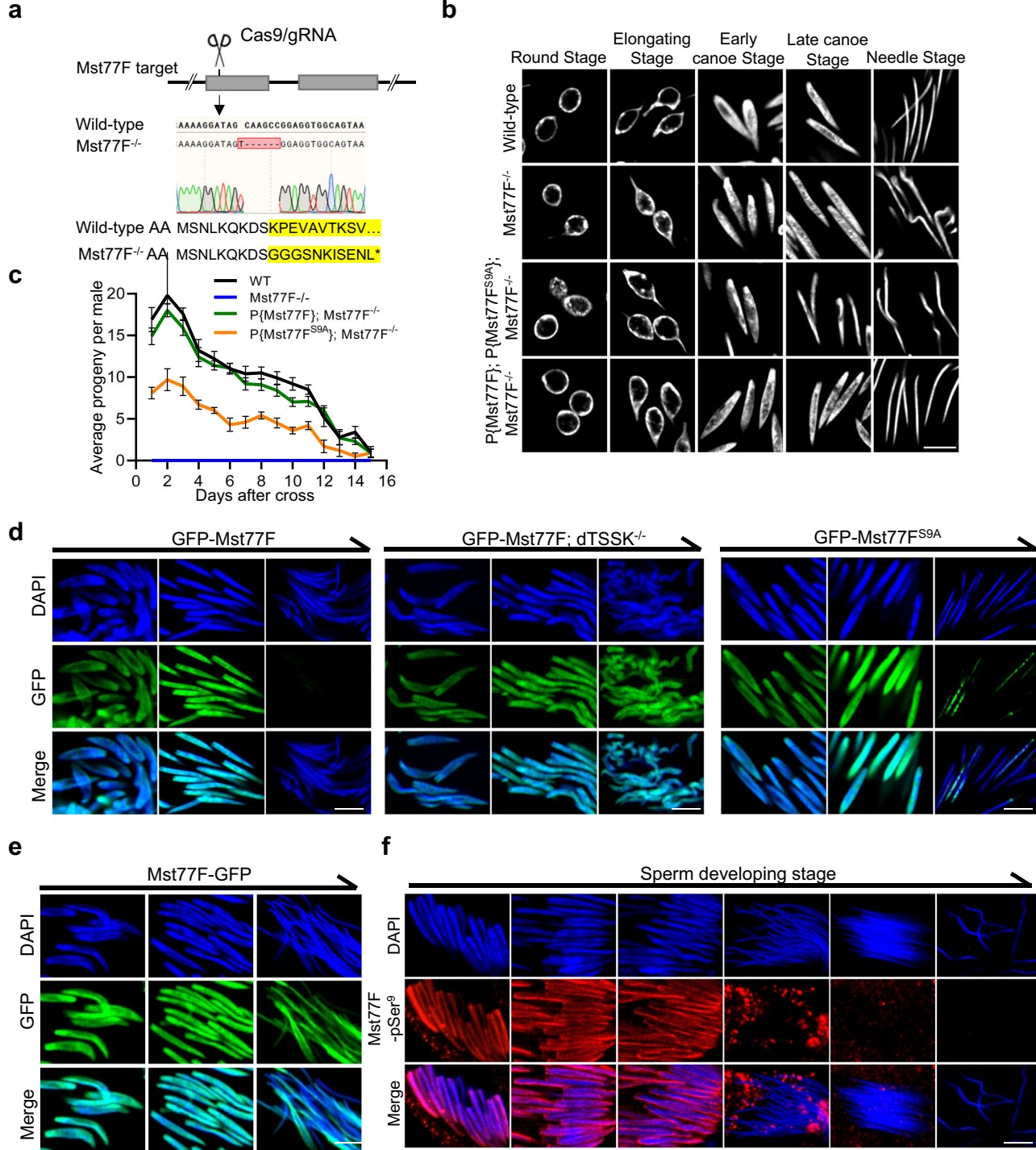

**Fig. 7 | Phosphorylation of Mst77F by dTSSK contributes to spermiogenesis.**
**a** CRISPR/Cas9-mediated Mst77F gene disruption in *Drosophila*. Sequencing results reveal a 1 bp insertion and 5 bp deletion in the coding region of the Mst77F gene, resulting in formation of a premature stop codon. **b** Sperm nuclear morphology at different stages of spermiogenesis in testes of wild-type, Mst77F⁻/⁻, and Mst77F- and Mst77F^S9A-rescued Mst77F⁻/⁻ flies. Disruption of the Mst77F gene results in severe defects of sperm DNA condensation. DNA stained with DAPI, white. Scale bar, 5 μm. **c** Qualitative fertility assay of wild-type (w¹¹¹⁸), Mst77F⁻/⁻, and Mst77F- and Mst77F^S9A-rescued Mst77F⁻/⁻ male flies (*n* = 10 per group). Disruption

of the Mst77F gene results in male sterility. Data are mean ± SEM. **d** Live imaging showing the localizations of Mst77F (GFP-Mst77F, green) and Mst77F^S9A (GFP-Mst77F^S9A, green) in spermatid nuclei at different stages of spermiogenesis (DNA stained with DAPI, blue). Scale bar, 5 μm. **e** Live imaging showing the localizations of Mst77F (Mst77F-GFP) in spermatid nuclei at different stages of sperm development (DNA stained with DAPI, blue; Mst77F-GFP, green). Scale bar, 5 μm. **f** IF showing the localization of Mst77F-pSer9 (red) at different stages of sperm development (DNA stained with DAPI, blue) in wild-type flies. Scale bar, 5 μm.

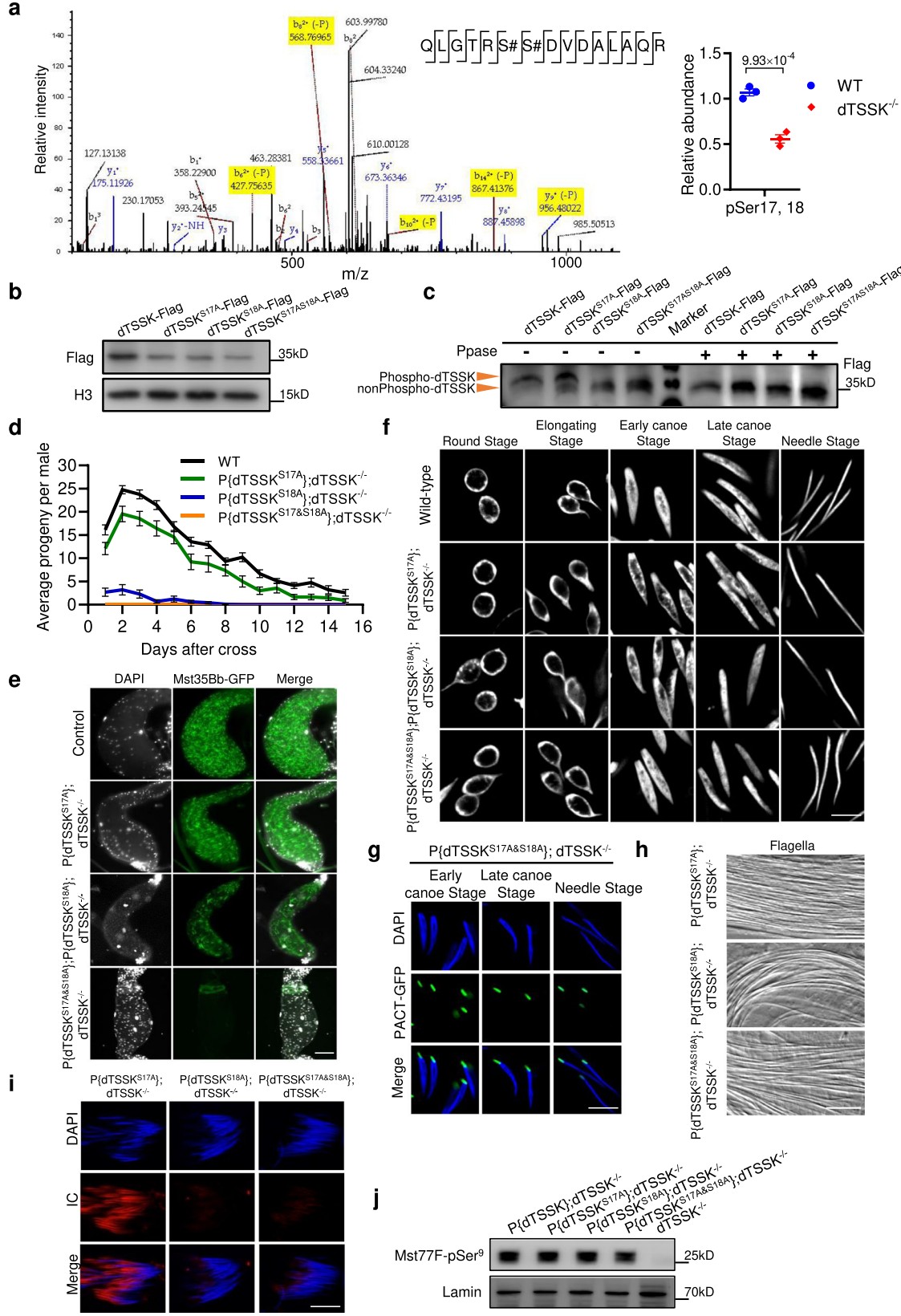

almost exclusively expressed postmeiotically in testes but exhibit different localization patterns during spermatid development[38]. Similar to mouse germ cells, all five TSSKs exhibit differential localization patterns during human spermiogenesis[38]. CG14305 (dTSSK), a homolog of mammalian TSSKs that we identified in *Drosophila*, was widely distributed in multiple key stages of sperm development, suggesting

that it is involved in multiple spermiogenesis functions. Multiple phenotypic defects such as sperm flagellar disorganization (a phenotype similar to that caused by *Tssk4* mutation in mice) and sperm chromatin decondensation (a phenotype similar to that caused by *Tssk6* mutation in mice) were observed during spermiogenesis[24,25]. The similarity of these phenotypic defects also implies the functional

**Fig. 8 | Autophosphorylation of dTSSK-Ser18 is essential for male fertility.**
**a** Detection of specific y (blue) and b (red) fragment ions allowed identification of the peptide sequence QLGTRSSDVDALAQR and assignment of a phosphorylation site to Ser17 and Ser18 of dTSSK. Plots (right) showing differences in the relative abundance of the phosphopeptide from dTSSK containing Ser17 and 18 between wild-type and dTSSK[−/−] flies (n = 3 per group). Data are mean ± SEM. Statistical analyses were performed by two-sided Student's t test. **b** WB analysis showing expression of dTSSK in testicular extracts of dTSSK-Flag, dTSSK[S17A]-Flag, dTSSK[S18A]-Flag, and dTSSK[S17A&S18A]-Flag flies. An anti-H3 antibody was used as a loading control. **c** WB analysis showing phosphorylation of dTSSK in testicular extracts of dTSSK-Flag, dTSSK[S17A]-Flag, dTSSK[S18A]-Flag, and dTSSK[S17A&S18A]-Flag flies. Ppase treatment removes the phosphorylation modification of dTSSK-Flag and dTSSK[S17A]-Flag. **d** Fertility assay of wild-type (n = 9), dTSSK[S17A] (n = 9), dTSSK[S18A] (n = 10), and dTSSK[S17A&S18A] flies (n = 10). Data are mean ± SEM. **e** Live imaging showing seminal vesicles of the indicated genotype stained with DAPI (white). Sperm nuclei (green) were labeled with Mst35Bb-GFP. Scale bar, 50 μm. **f** Sperm nuclear morphology at different stages of spermiogenesis in testes of wild-type, dTSSK[S17A], dTSSK[S18A], and dTSSK[S17A&S18A] flies (DNA stained with DAPI, white). Scale bar, 5 μm. **g** Live imaging showing the localization of the basal body at different stages of spermiogenesis in dTSSK[S17A&S18A] flies. PACT-GFP targeting the basal body, green. DNA stained with DAPI, blue. Scale bar, 5 μm. **h** Flagellar morphology in dTSSK[S17A], dTSSK[S18A], and dTSSK[S17A&18A] flies. Scale bar, 10 μm. **i** IF of phalloidin (red)- and DAPI (blue)-stained testes of dTSSK[S17A], dTSSK[S18A], and dTSSK[S17A&18A] flies showing the organization of ICs. Scale bar, 10 μm. **j** WB analysis showing Mst77F phosphorylation in testicular extracts of wild-type, dTSSK[S17A], dTSSK[S18A], and dTSSK[S17A&18A] flies. An anti-Mst77F-pSer[9] antibody was used to detect Mst77F phosphorylation. Lamin was used as a loading control. Source data are provided as Source Data file.

conservation of dTSSK with multiple mammalian TSSKs. Interestingly, in dTSSK mutant flies, the IC failed to form during spermiogenesis, which corresponded with the localization of dTSSK to the IC. Subsequently, rescue experiments confirmed that human TSSK1B (or TSSK1) and TSSK6 rescued the loose nuclear morphology of sperm in dTSSK mutants to a near mature needle-shape morphology, further demonstrating the functional conservation of dTSSK and mammalian TSSKs in chromatin condensation during spermiogenesis.

Previous studies on the function of TSSKs have consistently failed to establish whether the their crucial role in spermiogenesis is dependent on their kinase catalytic activity[21,24]. In this study, we did not detect phosphorylation at Mst77F-Ser9 in dTSSK[K60M] or dTSSK[T191A] flies, indicating that both sites are crucial for kinase activity of TSSK in vivo. More important, dTSSK[K60M] and dTSSK[T191A] flies displayed severe defects in spermiogenesis similar to those observed in dTSSK[−/−] flies. This demonstrates that the kinase catalytic activity of dTSSK is indispensable for its function in male fertility in vivo.

Structurally, the N- and C-terminal domains of dTSSK are not conserved with human TSSKs, which might confer some additional functions to dTSSK in spermiogenesis. Rescue experiments in *Drosophila* showed that sperm of flies expressing dTSSK that lacked the N- or C-terminal domain could develop to the needle stage, similar to human TSSK1B- and TSSK6-rescued flies, suggesting that the N- or C-terminal domain of dTSSK is not involved in compaction of sperm chromatin. Surprisingly, deletion of the N- or C-terminal domain of dTSSK led to an inability of sperm bundles to form normal IC structures (unique to *Drosophila* sperm maturation) and ultimately resulted in male infertility.

Taken together, our findings suggest that the conserved kinase catalytic domain S_TKc is critical for the function of TSSKs in spermiogenesis and for male fertility in various species from *Drosophila* to human.

## dTSSK regulates the histone-to-protamine transition

*Drosophila* is an excellent model to investigate sperm chromatin remodeling at the functional level because the histone-to-protamine transition in *Drosophila* shares conserved features with that in mammals[54]. A previous study has shown that the knockout of Tssk6 results in elevated levels of histones H3 and H4 in sperm[55]. We also found that dTSSK deficiency severely affected the histone-to-protamine transition. Although the protamine Mst35Bb was still incorporated into sperm chromatin in the late canoe stage, the histones H3 and H4 and the consequently loaded transition protein Tpl94D were not completely removed from sperm DNA in dTSSK[−/−] flies, resulting in incomplete incorporation of protamines due to the occupancy of these proteins on sperm DNA. Protamines are rich in bulky positively charged arginine residues (which mediate strong binding to the minor groove of negatively charged DNA) and cysteine residues (which form intra- and intermolecular disulfide bonds to stabilize DNA-protein binding)[56,57]; therefore, sperm protamine-DNA

assembles into an ultratight toroid[58], resulting in at least sixfold denser nuclear compaction compared with histone-bound DNA[59]. The failure of histones to be completely removed from sperm DNA would further impede sperm DNA compaction, which is perhaps related to the inability of dTSSK[−/−] sperm to transform into a needle shape. Unfortunately, we did not detect obvious phosphorylation modifications of H3, H4, and Tpl94D in dTSSK[−/−] and wild-type flies in our phosphoproteomic analysis. dTSSK may not directly phosphorylate histones and transition proteins but rather regulate the histone-to-protamine transition in an indirect manner, such as by phosphorylating and thereby activating ubiquitin-modifying enzymes. It remains unknown which dTSSK-regulated factors are required for histone eviction/protamine replacement during spermiogenesis.

## Multiple potential substrates phosphorylated by dTSSK are involved in spermiogenesis

Our phosphoproteome analysis may provide some explanation for how dTSSK mutation causes various phenotypic defects. The nuclear morphology of the sperm head is mainly determined by two factors: nuclear shaping and chromatin condensation[29]. Nuclear shaping is driven by microtubules that emanate from the basal body and associate with the nuclear envelope. Interestingly, microtubule-based genes such as wampa, Grip75, and Grip91 are reportedly involved in microtubule nucleation and nuclear shaping during spermiogenesis[60–62]. For chromatin condensation, Mst77F is required for compaction of sperm DNA[31]. Our findings suggest that dTSSK has the potential to regulate the phosphorylation of Mst77F both in vitro and in vivo. More importantly, we discovered that proteolysis of the Mst77F N-terminal domain depended on dTSSK-mediated phosphorylation of Mst77F-Ser9, which probably explains why the Mst77F[S9A] mutation resulted in production of spermatozoa with an abnormal nuclear morphology. Interestingly, N-terminal proteolysis of protamine P2 in mammals is reportedly involved in spermiogenesis and is associated with some forms of human infertility[63,64]. Thus, we propose that a similar mechanism might regulate sperm DNA condensation in mammals. It is also possible that other potential substrates of dTSSK such as Mst33A co-operate to regulate sperm chromatin packaging. Unfortunately, we could not identify mammalian homologues of Mst33A using sequence blast searches.

In addition, disorganized assembly of flagella could affect spermatid maturation processes such as motility. Given that centriole deficiency always leads to flagellar defects[65], and dTSSK mutation results in degeneration of basal body localization on sperm, some potential centriole-associated substrates such as Cp110, Spd-2, and CG3213 (Odf2 homolog in humans) might be involved in assembly of cilia and flagella via phosphorylation by dTSSK[28,66,67]. Other potential flagellate-associated substrates, kl-3, TTLL3B, Orb2, and Rsph3, are involved in the composition of sperm axonemes[50,68–70], and kl-3 and TTLL3B deficiency causes male sterility[68,71]. Moreover, other potential substrates such as didum, Lasp, Pif1A, and klhl10 are involved in the

individualization process[53,72–74]. Among them, human CCDC157 (Pif1A homolog in flies) and klhl10 are relevant to nonobstructive azoospermia[74,75]. Our phosphoproteomic analysis also identified other male infertility-associated genes, such as ms(3)76 Cc, cmb, and Fas3[50,52,76], whose regulation by dTSSK during spermiogenesis warrants further investigation.

Furthermore, it is noteworthy that some substrate proteins identified in our study by phosphoproteomics as being regulated by *Drosophila* dTSSK exhibit homology with proteins that are regulated by Tssk3 in mice[23], such as kl-3 (Dnah8 homolog in mice), Osbp2 (Osbp homolog in mice), Ref. ($^2$)p (Sqstm1 homolog in mice), didum (Radil homolog in mice), Sls (Ttn homolog in mice), and TTLL4B (Ttll5 homolog in mice). These findings suggest that the molecular mechanisms underlying spermiogenesis may be conserved across species, and further research into the regulation of these homologous substrate proteins by TSSKs could provide valuable insights into the evolution and function of these critical signaling pathways in male fertility.

### Clinical relevance of TSSKs

Our findings revealed that dTSSK could organize and orchestrate the entire postmeiotic spermiogenesis process by phosphorylating multiple protein substrates involved in different stages of germ cell development, providing valuable and paradigm-shifting insights into spermiogenesis. Future work should aim to confirm the potential targets of dTSSK and identify the corresponding targets of human TSSKs, to further elucidate the molecular mechanisms underlying human infertility. Indeed, single-nucleotide mutations of TSSK2, TSSK4, and TSSK6 are reportedly associated with azoospermia and severe oligospermia[77–79]. In addition, members of the TSSK family such as TSSK2, TSSK3, and TSSK6 play important roles in survival and proliferation of cancer cells, and TSSK6 is an immunogenic cancer/testis antigen in various cancers[80–83]. Our discovery of the broad-spectrum but selective phosphorylation of substrates by TSSKs enhances understanding of clinical human male infertility and tumorigenesis. Furthermore, TSSKs have been suggested as promising targets for male contraception[20]. Our current study provides further mechanistic evidence that TSSKs play a critical role in sperm development and maturation, and their inhibition could disrupt spermiogenesis and impede the production of functionally competent sperm. It is worth noting that although promising progress has been made in developing contraceptive drugs using TSSK inhibitors in animals, further research is needed to establish their efficacy and safety in humans.

## Methods

### Generation of transgenic lines

To generate P{Flag-GFP-dTSSK} transgenic flies, the endogenous promoter was amplified from *w^1118* genomic DNA with the primers CG14305-pro-F/CG14305-pro-R and cloned into the *pUAST-GFP-attB* vector between the *NotI* and *AgeI* sites. The genomic region covering the CG14305 gene locus (from ATG to 1 kb downstream of the coding region) was amplified from *w^1118* genomic DNA using the primers CG14305-F/CG14305-R. The PCR fragment was cloned between the *SpeI* and *KpnI* sites of the *pUAST-GFP-attB* vector. The plasmids were injected into *attP40* embryos for targeted phiC31-mediated integration at genomic attP landing sites. To generate P{dTSSK} transgenic flies, the full CG14305 gene locus was amplified with the primers CG14305-pro-F/CG14305-R and cloned into the *pUAST-attB* vector between the *NotI* and *KpnI* sites. To generate P{PACT-mCherry} and P{PACT-GFP} transgenic flies, the C-terminal 226 amino acids of CG6735 (which contains the PACT domain) were amplified by PCR from *Drosophila* testicular cDNA as described previously[40] with the primers PACT-F/PACT-R. The *ubi* promoter and 3′UTR (about 1 kb downstream of the coding region) were amplified from *w^1118* genomic DNA using the primers ubi-F/ubi-R and PACT-3′

UTR-F/PACT-3′UTR-R. mCherry and GFP were amplified from the *pHPdestmCherry* (Addgene #24567) and *pUAST-GFP-attB* vectors using the primers mCherry-F/mCherry-R and GFP-F/GFP-R. The amplified products were cloned step by step into the *pUAST-attB* vector. To generate P{Mst35Bb-mCherry} and P{Mst35Bb-GFP} transgenic flies, the genomic region covering the Mst35Bb promoter and gene body was amplified using the primers Mst35Bb-F/Mst35Bb-R, and the 3′UTR was amplified using the primers Mst35Bb-3′UTR-F/Mst35Bb-3′UTR-R. Finally, the mCherry and GFP fragments were respectively cloned into the preconstructed *pUAST-attB* vector. To generate P{H2A-GFP}, P{H2B-GFP}, P{H3-GFP}, P{H4-GFP}, P{H2Av-GFP}, and P{H3.3-GFP} transgenic flies, the promoter and 3′UTR of H3.3 A were used to drive all histone expression in testes and cloned into the *pUAST-GFP-attB* vector. H2A, H2B, H3, H4, H2Av, and H3.3 A were respectively amplified from cDNA and cloned into the vector. To generate P{Tpl94D-GFP} transgenic flies, the Tpl94D promoter, coding region, and 3′UTR were amplified from *w^1118* genomic DNA and cloned into the *pUAST-GFP-attB* vector. Similarly, to generate P{Flag-GFP-Mst77F}, P{Mst77F-GFP}, and P{Flag-GFP-Mst33A} transgenic flies, the relevant promoter, coding region, and 3′UTR were amplified from *w^1118* genomic DNA and cloned into the *pUAST-attB* vector. To generate five human TSSK transgenic flies, the CG14305 endogenous promoter amplified from *w^1118* genomic DNA and human TSSK sequences synthesized by GenScript were respectively cloned into the *pUAST-attB* vector. To generate point mutants of dTSSK, Mst77F, and Mst33A transgenic flies, the gene coding region was amplified and cloned into the T vector (Transgene #CT101-01). Point mutation primers were used to amplify sequences from the constructed T vector. The mutation fragments were cloned into the *pUAST-attB* vector. P{Mst33A}, P{Mst33A^S237A}, and P{Flag-GFP-Mst33A^S237A} plasmids were injected into *attP2* embryos, and other transgenic plasmids were injected into *attP40* embryos for targeted phiC31-mediated integration. All primers used to generate transgenic flies are listed in Supplementary Data 3. Flies used in this study are listed in Supplementary Data 4.

### Generation of *Drosophila* mutants

Mutants of dTSSK^−/−, Mst77F^−/−, and Mst33A^−/− flies were generated by CRISPR/Cas9-mediated mutagenesis as previously described[84]. Briefly, in vitro transcription of Cas9 mRNA was performed using an Sp6 mMESSAGE mMACHINE Kit (Thermo Fisher Scientific #AM1340). In vitro transcription of the designed gRNAs was performed using a RiboMAX Large Scale RNA Production Systems-T7 Kit (Promega #PR-P1320). Purified Cas9-mRNA and gene-specific gRNAs were mixed at final concentrations of 1 μg/μL and 50 ng/μL, respectively, and injected into *w^1118* embryos. The mutants were verified by PCR and DNA sequencing. Primers used for gRNA construction, PCR, and sequencing are listed in Supplementary Data 3.

### Molecular cloning and S2 cell transfection

To generate the Flag-Mst77F plasmid, the Mst77F coding region was amplified from a homemade *Drosophila* testis cDNA library using the primers Pac-Mst77F-F/Pac-Mst77F-R and cloned into the pAC5.1/V5-His vector between the *XbaI* and *AgeI* sites for expression in *Drosophila* S2 cells (Thermo Fisher Scientific, R69007). dTSSK-V5 was amplified from the *Drosophila* testis cDNA library using the primers Pac-dTSSK-F/Pac-dTSSK-R and cloned into the pAC5.1/V5-His vector between the *KpnI* and *XhoI* sites. The primer sequences are listed in Supplementary Data 3. The plasmids used for transfection were prepared using a Plasmid Mini Kit (Tiangen #DP118) and transferred into cells using transfection reagents (Qiagen #301425). To improve efficiency, it is advisable to split the culture 1:1 with fresh medium 1 day before transfection. Transfected cells were allowed to grow for 72 h at 25 °C and were harvested via centrifugation at 3000 × *g* for 5 min.

## Antibody generation

Two antibodies, Mst77F-pSer9 and dTSSK, were generated by Abclonal, a company based in Wuhan, China. The Mst77F-pSer9 antibody was produced using a modified peptide composed of amino acids 5-13: KQKD(S-p)KPEV, while the dTSSK antibody was produced using a peptide consisting of amino acids 14-21: GTRSSDVD.

## Sequence alignment and phylogenetic analysis

Protein sequences were aligned using ClustalW of Mega software[85]. Identity and similarity were calculated using Blastp[86]. The phylogenetic tree was constructed using Phylogenetic Analysis of Mega software[87].

## Fertility testing

Every male or virgin female of different fly mutants was crossed with three $w^{1118}$ virgin females or males. Parents were kept for 7 days and discarded. Then, adult flies of each cross were counted to assess fertility of the detected fly mutants. For the qualitative fertility assay, males were tested in batches of ten. Each virgin male was placed with one $w^{1118}$ virgin female in one vial at 25 °C. For the next 15 days, the flies generated from each mating were transferred to new vials every 24 h. Upon eclosion, all progeny from each vial were counted. The average number of flies per parental pair and standard errors were calculated for each combination of genotypes.

## IF and microscopy

For whole testis imaging, testes of 3 days after eclosion were dissected in phosphate-buffered saline (PBS) and transferred to a small drop of PBS on a microscope slide. Then, a coverslip was gently placed on the slide for cytological examination. For standard sperm imaging, testes of 3 days after eclosion were dissected in PBS and transferred to a small drop of PBS on a microscope slide. Then, a coverslip was placed on the slide to release the contents of testes. For IF sample preparation, 20 pairs of testes were dissected in cold PBS. Next, the sample was transferred to a 1.5 mL centrifuge tube and incubated in 4% PFA for 20 min. After three rinses for 10 min with 1× PBST (1× PBS containing 0.1% Triton X-100) at RT, samples were blocked for 1 h in 1× PBST containing 1% BSA (Gibco). Primary antibody staining was performed overnight in block buffer at 4 °C. After three washes with 1× PBST, the secondary antibody and DAPI diluted in 1× PBST were added for 1 h at RT. Goat anti-Mouse IgG (H + L) Highly Cross-Adsorbed Secondary Antibody, Alexa Fluor™ 488 (Invitrogen #A-11029) or Alexa Fluor™ 568 (Invitrogen #A-11036) were used at a dilution of 1:500. For IC staining, samples were stained with TRITC phalloidin as previously described[41]. Briefly, ten pairs of testes were dissected in cold PBS, incubated in sodium isocitrate for 5 min, and then incubated in 4% PFA for 20 min. Tissue samples were flattened under a siliconized coverslip and frozen in liquid nitrogen, and then the coverslip was removed. The microscope slides were incubated in cold 1× PBST for 10 min and then in RT 1× PBST for 10 min, blocked in PBSTA (1× PBS containing 0.1% Triton X-100 and 5% BSA), incubated in 1× PBSTA containing TRITC phalloidin (Yeasen #40734ES75) for 1 h at 37 °C, washed with 1× PBST, and stained with DAPI. Confocal images were obtained using a Leica SP8 or Zeiss LSM 980 confocal microscope. Images were processed with Fiji software[88].

Quantification of fluorescence intensity of GFP in the sperm was performed using Image J. The area and integrated optical density in sperm were measured respectively to obtain mean signal intensity.

## SDS-PAGE, Phos-tag SDS-PAGE, and WB analysis

For total protein extraction, transfected cells or fly tissues were suspended in 1.5× SDS loading buffer. After brief sonication, the samples were boiled for 10 min at 95 °C, centrifuged at 25,000 × $g$ for 20 min, and loaded for SDS-PAGE according to the manufacturer's protocol. Proteins were transferred to a PVDF membrane in transfer buffer at 25 V for 25 min using the Trans-Blot® Turbo™ Transfer System (Bio-

Rad). After blocking with 5% BSA diluted in PBSTween (PBS containing 0.1% Tween 20) for 2 h at RT, membranes were incubated overnight at 4 °C with a primary antibody diluted in PBSTween containing 5% dry milk. The following day, blots were washed three times with PBSTween for 10 min, incubated with a secondary antibody diluted in PBSTween for 1 h at RT with shaking, and then washed again with PBSTween. Membranes were then incubated with peroxidase-conjugated secondary antibodies diluted in PBSTween for 1 h and washed three times with PBSTween for 10 min at RT.

For $Mn^{2+}$-Phos-tag SDS-PAGE, 40 μM $MnCl_2$ and 20 μM acrylamide-pendant Phos-tag ligands were added to a 12% separating gel before polymerization. Testicular samples were suspended in 1.5× SDS loading buffer. After brief sonication, the samples were boiled for 10 min at 95 °C, centrifuged at 25,000 × $g$ at 25 °C for 20 min, loaded for Phos-tag SDS-PAGE, and electrophoresed at 80 V. After electrophoresis, the gel was immersed in transfer buffer (39 mM glycine, 48 mM Tris, 0.037% SDS, and 20% methanol) containing 10 mmol/L EDTA and gently shaken for 10 min (this was repeated three times). Then, the gel was soaked in EDTA-free transfer buffer and shaken gently for 10 min. The subsequent procedures were performed using standard protocols. For Ppase processing, samples were lysed in NEBuffer for Protein MetalloPhosphatases containing 1 mM $MnCl_2$ and incubated with Ppase (New England Biolabs #P0753) for 30 min at 30 °C. WB analysis was performed using standard procedures. A mouse polyclonal anti-lamin antibody (DSHB #LC28.26) was used at a 1:1000 dilution, a rabbit polyclonal anti-Histone H3 antibody (Abcam #ab1791) was used at a 1:1000 dilution, a mouse polyclonal anti-V5 antibody (Invitrogen # MA5-15253) was used at a 1:1000 dilution, a rabbit polyclonal anti-Flag antibody (Millipore #F7425) was used at a 1:1000 dilution, a rabbit polyclonal anti-dTSSK antibody (ABclonal) was used at a 1:2000 dilution, and a rabbit polyclonal anti-Mst77F-pSer9 antibody (ABclonal) was used at a 1:5000 dilution. Peroxidase-conjugated anti-mouse or anti-rabbit secondary antibodies (Yeasen #33201ES60 or #33101ES60) were used at a 1:5000 dilution. WB signals were detected with a GE AI680UV instrument and processed with Fiji software[88].

## Peptides

The peptides of Mst77F-S9: "KQKDSKPEV", Mst77F-S9A: "KQKDAKPEV", Mst77F-PS9: "KQKD(S-p) KPEV", N o-S/T-Control: "DIEEDINRAEDE", Mst33A-S237: "NIIYSGSAYKNFL" and dTSSK peptide: "GTRSSDVD" are synthesized by GenScript.

## In vitro kinase assay

For sample preparation, 100 pairs of testes of Flag-dTSSK-expressing flies were dissected in cold PBS and centrifuged at 3000 × $g$ for 5 min at 4 °C. After discarding the supernatant, 600 μL of RIPA buffer (50 mM Tris-HCl, pH 7.4, 150 mM NaCl, 1% NP-40, 0.5% sodium deoxycholate, and 0.1% SDS) containing 1× protease inhibitor cocktail (Sigma), 1 mM PMSF, 1 mM NaF, 1 mM DTT, 0.5 mM $Na_3VO_4$, 0.5 mM EDTA, and 0.5 mM EGTA was added, and samples were briefly sonicated on ice. After centrifugation at 25,000 × $g$ for 30 min at 4 °C, an equal amount of Flag M2 gel (Sigma #A2220) was added to the supernatant, and samples were incubated for 5 h at 4 °C. After three washes with 1× Tris-buffered saline (TBS), the precipitate containing beads was collected by centrifugation at 250 × $g$ for 1 min at 4 °C. The precipitates were incubated with synthetic peptides diluted in kinase buffer (25 mM HEPES, 10 mM $MgCl_2$, and 0.5 mM EGTA) containing 1× protease inhibitor cocktail (Roche), 1 mM PMSF, 1 mM NaF, 1 mM DTT, 0.5 mM $Na_3VO_4$, 0.5 mM EDTA, and 0.5 mM EGTA for 30 min at 25 °C.

The supernatants were blotted onto nitrocellulose membrane. After drying, the membranes were blocked by incubation with TBS containing 0.1% Tween-20 (0.1% TBSTween) and 5% skim milk for 2 h. Blots were washed with TBST and then incubated overnight at 4 °C with an anti-Mst77F-pSer9 or antiphosphoserine antibody (Merck #05-

1000) was used at a 1:1000 dilution. After three washes with 0.1% TBSTween, the membranes were incubated with a peroxidase-conjugated secondary antibody for 1 h at RT. The membranes were washed again three times with 0.1% TBSTween and then treated with a chemiluminescent reagent. Signals were detected with a GE AI680UV instrument. Images were processed with Fiji software.

## Quantitative phosphoproteomics

About 1000 pairs of testes from dTSSK$^{-/-}$ and wild-type flies of 3 days after eclosion were collected and stored at −80 °C. Frozen samples were lysed in lysis buffer (8 M urea, 1% SDS, 1 mM EDTA, 1 mM PMSF, 10 mM NaF, and phosphatase inhibitor cocktail [1:100 dilution]) by disruption three times for 40 s using Tissuelyser (Jingxin). The lysed tissue was placed on ice for 10 min and vortexed for 10 s. This process was repeated twice. Cell debris was removed by centrifugation at 16,000 × $g$ for 30 min at 4 °C. The protein-containing supernatant was collected, and the protein concentration was measured using the BCA assay (Pierce). For each sample, 300 μg of protein was used for downstream reduction, alkylation, and digestion. Triethylammonium bicarbonate buffer (TEAB) was added to each sample at a final concentration of 100 mM. Proteins were reduced and alkylated with tris (2-carboxyethyl) phosphine hydrochloride (Thermo Fisher; 10 mM, 37 °C, 1 h) and iodoacetamide (Sigma; 40 mM, RT in the dark, 40 min), respectively. Six volumes of precooled acetone were added to the sample, and proteins were precipitated at −20 °C for 4 h. Protein precipitates were collected by centrifugation at 10,000 × $g$ for 20 min at 4 °C and completely dissolved in 100 μL of 100 mM TEAB. The protein samples were added to trypsin (Promega) at a ratio of 50:1 (protein: enzyme, w/w) and incubated at 16 h. Peptides were dried using Speed-Vac (Thermo Scientific), dissolved in 0.1% trifluoroacetic acid containing 2% acetonitrile, desalted on HLB columns (Waters), and dried again using Speed-Vac (Thermo Scientific).

Dried peptides from each sample were chemically labeled with one of six TMT Isobaric Mass Tags (Thermo Fisher Scientific #90111). TMT reagents were dissolved in anhydrous acetonitrile (Sigma), and each reagent was added to the corresponding aliquot of peptides. The reaction was incubated at RT for 2 h and quenched by incubation with hydroxylamine at RT for 30 min. Equal amounts of labeled peptides from each group were mixed in one tube. The labeled peptides were desalted on HLB columns (Waters) and dried using Speed-Vac (Thermo Scientific). Phosphorylated peptides were enriched using a High-Select TM Fe-NTA Phosphopeptide Enrichment Kit (Thermo Fisher Scientific #A32992) following the instruction manual, desalted using C18 columns (Waters), and dried using Speed-Vac (Thermo Scientific). Dried phosphopeptide samples were resuspended in 0.1% formic acid for LC-MS analysis. Phosphopeptides were separated on a reverse phase C18 column (75 μm × 25 cm, Thermo) over 180 min with an acetonitrile gradient from 20 to 80%. Then, samples were electrosprayed and analyzed by tandem mass spectrometry on a Q Exactive HF-X instrument (Thermo Fisher Scientific). The parameters for phosphoproteomic samples were set to a mass range of 350–1500 m/z; resolution of 120,000, automatic gain control (AGC) target of 3.0e6, and maximum injection time of 50 ms for MS1; and resolution of 45,000, AGC target of 1.0e5, and maximum injection time of 100 ms for MS2.

Raw mass spectrometry data were analyzed using ProteomeDiscoverer™ Software 2.4. MS/MS spectra were matched against UniProt-*Drosophila melanogaster*-taxonomy 7227. Identifications were filtered at 1% FDR at the peptide level, accepting a minimum peptide length of 5. TMT intensities were extracted, normalized for each condition, and used for downstream analyses. Normalized TMT intensities were analyzed using the Student's $t$ test (two-tailed, homoscedastic), and phosphopeptides with $P < 0.05$ and fold change >1.5 were considered to significantly differ in abundance between the samples. Significant downregulated mass spectrometry hits were used for functional enrichment analysis in g:Profiler[89] (https://biit.cs.ut.ee/gprofiler).

## Motif and network analysis

Significant mass spectrometry hits (fold change >1.5, $P < 0.05$) were analyzed using UniProt, motif-x, and the STRING database. Motif-x software (v1.2 10.05.06)[90] was used with prealigned sequence tags around the phosphosites. The following parameters were applied: occurrences, 10; significance, $P < 0.0001$. The STRING database v11.5 was used[91]. Analysis was performed using default parameters (medium confidence). Network parameters, including the source and target node defining the direction of the interaction and the combined interaction score defining the weight of the network edge, were imported into Gephi v0.9.2[92] for network visualization. Networks were visualized as ForceAtlas2 with edge thickness proportional to the combined interaction score.

## Proteomics

About 500 pairs of testes from dTSSK$^{-/-}$ and wild-type flies of 3 days after eclosion were collected and stored at −80 °C. Frozen samples were lysed in lysis buffer (8 M urea, 50 mM ammonium bicarbonate, 1 mM dithiothreitol) with protease inhibitor added (10 mL/tablet, 05892791001, Roche) by disruption three times for 40 s using Tissuelyser (Jingxin), respectively. The suspension was sonicated on ice for 10 s, repeated thrice. The protein concentration of the lysates was determined using the bicinchoninic acid (BCA) assay. About 1 mg of lysate was then reduced with 5 mM dithiothreitol (DTT) at 37 °C for 2 h, and alkylated with 20 mM iodoacetamide for 45 min in the dark. After diluting the concentration of urea to 1 M with 50 mM ammonium bicarbonate (ABC), trypsin (V5111, Promega) was added at a weight ratio of 1:50 and incubated for 16 h at 37 °C. The reactions were stopped by adding 20% formic acid (FA) until the pH was less than 2. The resulting peptides were desalted using Sep-Pak C18 (WAT054955, Waters) according to the manufacturer's instructions and then dried under vacuum. Dried peptides were resuspended in 0.1% formic acid for LC-MS analysis. Peptides were separated and analyzed on an Easy-nLC 1000 system coupled to a Q Exactive HF (Thermo Scientific). The raw data were processed using MaxQuant with integrated Andromeda search engine (v.1.5.4.1). The UniProt *Drosophila melanogaster*-taxonomy 7227 protein database were used for database searches of samples. False-discovery rate (FDR) thresholds for peptide was specified at 1%. Minimum peptide length was set at 7. A paired Student's $t$ test was used to verify the significance of the differences between each comparison.

## Statistics and reproducibility

Immunofluorescence and western blotting results were repeated at least twice using independently prepared samples, and similar results were obtained. The statistical methods used are indicated in the legends. Statistical analysis was performed using GraphPad Prism version 8.0. Sample size was not predetermined by any statistical method.

## Reporting summary

Further information on research design is available in the Nature Portfolio Reporting Summary linked to this article.

## Data availability

The mass spectrometry proteomics data have been deposited to the ProteomeXchange Consortium (http://www.proteomexchange.org/) via the PRIDE partner repository with the dataset identifier PXD038767 and PXD041466. The remaining data are available within the paper, Supplementary Information or Source Data file. Source data are provided with this paper.

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

## Acknowledgements

We thank Dr. Chao-Po Lin, Dr. Jilong Liu and Dr. Gaofeng Fan for helpful discussions. This work was supported by grants from the National Nat-ural Science Foundation of China (32070846 and 91740107 to G.G.), National Postdoctoral Fellowship (2019M661652 to X.Z.), and Science and Technology Commission of Shanghai Municipality (19JC1413600 to G.G.). We thank the Molecular Imaging Core Facility (MICF) and Mole-cular and Cell Biology Core Facility (MCBCF) at the School of Life Sci-ence and Technology, ShanghaiTech University, for providing technical support. Additionally, we also thank the staff members of the Mass Spectrometry team of ShanghaiTech University for technical support with the LC-MS/MS experiment.

## Author contributions

G.G. designed the whole project. Z.X., P.J., and W.M. generated the mutants and transgenic flies. P.J. and S.A. performed the fertility testing and testicular imaging. Z.X., P.J., S.W., and Z.J. collected testes for phosphoproteomic analysis. Z.X., P.J., and Z.J performed biochemical experiments. W.X. and Z.X performed bioinformatics analysis. G.G., Z.X., P.J., and W.M. wrote the paper. All authors discussed the results and declared that there was no competing interests.

## Competing interests

The authors declare no competing interests.
