## [Peer Review File · Nature Communications]

Broad phosphorylation mediated by testis-specific serine/threonine kinases contributes to spermiogenesis and male fertilityREVIEWER COMMENTS

Reviewer #1 (Remarks to the Author):

This work by Zhang et al. explores the functional role of a member of the testis-specific serine kinase family known as TSSKs in flies. These kinases, originally identified in mice have been shown bioinformatically to be present in many other organisms including *C. elegans* and *D. melanogaster*. In this manuscript, the authors focusing in the investigation of the only TSSK found in flies which was named dTSSK. Overall, this is an outstanding study and the first one to use advanced genetic tools to investigate the role of dTSSK, in particular, and TSSKs overall in spermiogenesis. As found with the *Tssk1/Tssk2* double KO, and the single TSSK3 and TSSK6 models, flies lacking dTSSK are sterile. However, this work is able to continue functional studies by conducting rescue experiments with wild type dTSSKs as well with different other genes, including all the human orthologues and several mutants. Overall, this is an outstanding work that transcends the *Drosophila* field and can be used to understand what TSSKs are doing in other systems. More generally, it also offers tools to investigate one of the less known aspects of spermatogenesis which is spermiogenesis.

Despite these positive comments, I have found several aspects that need revision. First, it is surprising the lack of comments on papers describing loss-of-function knock out genetic models in the mouse (see relevant references below). Second, Western blots are shown with cut bands. Third, some of the statements are over-conclusions that deserved to be toned down. Fourth, although the authors have conducted a very good phospho proteomic analysis, addition of a comparative standard proteomic analysis would enhance the impact of this work. Finally, it is important throughout the paper to recognize that decrease in phosphorylation sites does not mean direct phosphorylation by dTSSK. The authors should consider the involvement of kinase cascades in more complex signaling pathways.

Specific Comments.

- The authors have cited most works on the field. It is therefore surprising that they did not cite those showing genetic loss-of-function mice models indicating the essential nature of TSSK1, 2 (1)(double KO), 3 (2, 3) and 6 (4) as well as subfertility in the case of *Tssk4* KO (5).
- Fig. 1 A and Fig. 1 E. Complete blots should be shown including MW.
- The rescue control with wild type dTSSK is an excellent approach and prepares the reader for the rest of the work.
- Line 184. Although in humans there are only five TSSK genes, in other species such as the mouse, *Tssk5* is also present. So, instead of saying the five human *Tssk* genes were used for rescue, mention that these genes are 1, 2, 3, 4 and 6. (MINOR).
- Line 189. Analyses of nuclear morphology and histone fate was done with TSSK1 B and TSSK6-rescued flies but not with the others. The authors should make a comment about why TSSK2-, TSSK3- and TSSK4- rescued flies were not further analyzed.
- Line 207. I agree with the authors that the experiments described here are good. However, I believe that description of this particular experiment should use "strongly suggest" instead of ":demonstrate". (MINOR)
- Line 216. This sentence is not accurate. I agree with the conclusion but not with the premise. From this reviewer point of view what suggest that the N- and C- terminal-rescued flies are essential for sperm maturation is the fact that these flies can progress to a later spermiogenesis stage upon expression of dTSSK Δ N, and dTSSK Δ C.

- Although difficult to conduct, instead of a gain-of-function rescue, a loss-of-function of dTSSKΔN, and dTSSKΔC will be more conclusive regarding the relevance of these fragments.
- Line 267 to 268. A sentence stating if these identified proteins have mammalian homologues will be helpful for those readers that do not know Drosophila nomenclature. (MINOR)
- Line 277. Please change "To demonstrate". Instead use "To evaluate or To test".
- Line 280. It is important that the authors state here (and for the general phosphoproteomic data) that the lack of phosphorylation of a particular protein does not imply that this protein is a dTSSK substrate. It only indicates that dTSSK is part of the pathway. The MAPK pathway is an excellent example to visualize the problem with this conclusion.
- Line 283. The Western blot reveals absence of the phosphorylation site. However, in the absence of a control blot with an antibody that recognize total Mst77F, the authors cannot claim that the lack of signal is due to lack of phosphorylation or to reduced Mst77F protein expression. Ideally, this control should be added. In Fig. 6 E, this point is addressed by heterologous expression of a tagged Mst77F. So, overall, this comment is MINOR.
- Ideally, a proteomic analysis should be added to complement the results. It is not clear if the phospho peptides observed are due to decreased phosphorylation or decreased protein expression. Addition of a standard comparative proteomic analysis would be informative of the role of dTSSK in controlling protein synthesis and will increase the impact of this work.
- Line 308. This conclusion is not accurate. Don't take me wrong, these experiments are excellent; however, they do not "demonstrate" direct phosphorylation. The co-expression experiment confirms that when Mst77F is present dTSSK mediates its phosphorylation through an unknown pathway. It is silent regarding direct phosphorylation because other kinases are present. Demonstration of direct phosphorylation would require an in vitro assay with purified protein. However, even this experiment would be not completely conclusive because it will not be possible to state that direct phosphorylation occurs in vivo. In line 348, a more direct experiment is described. The authors, here, immunoprecipitated the endogenous dTSSK from cell-free extracts and conduct an in vitro kinase assay. Although this experiment limits the amount of proteins associated to dTSSK, still cannot discard that dTSSK IP brings down other associated kinases responsible for this phosphorylation.
- Line 344. It is important that the authors try to describe mammalian orthologues (if available).
- Line 365. Please change the word "determine" to "evaluate/test/investigate" (or another word that does not imply that the authors expect their hypothesis to be correct).
- Fig.8 C and D. Add whole Western blots in a supplementary figure.
- Line 376. Line 365. Please change the word "demonstrate" to "evaluate/test/investigate". (or another word that does not imply an expected result).
- Line 403. This expression is correct. The authors should use it throughout the paper. The word "potential" here is essential.
- Line 411. Citation needed for the mammalian case.
- Line 415: Not clear why the authors said that dTSSK function is distinct from the functional role of TSSKs in mammals. First, there are 6 TSSKs in the mouse and five in humans, their role in spermiogenesis is not well defined, so, it is difficult to state that they don't have similar functions. A plausible hypothesis is that the dTSSK role divided in mammals require more than one TSSK. Second,

although rescue is not complete, results in the present work showed that, at least TSSK1B and TSSK6 are able to rescue many defects observed in the dTSSK KO flies. Third, it is not possible to discard that expression of a combination of mammalian TSSKs would result in complete rescue of the fertility phenotype in flies.

- Line 425. This sentence "In vitro experiments consistently fail..." is not clear. Which in vitro experiments, the authors are referring to? Are they referring to the IP of dTSSKs from flies? Please explain in the text.

- Line 446. It is very important here to cite Jha et al. JBC paper showing that in the absence of TSSK6, there are problems with histone replacement, reminiscent of the findings in the present work (6).

- Line 455. In the TSSK6 KO model, H2AX (a phosphorylated form of H2) is not found during spermiogenesis. However, in vitro experiments show that there is no direct phosphorylation suggesting that this phosphorylation is part of a more complex kinase cascade.

- The hypothesis of phosphorylation of ubiquitin-modifying enzymes is attractive.

- As a final comment, many authors have considered TSSKs as targets for contraception (7). A couple of sentences mentioning this point will be of interest for many readers.

- A previous phosphoproteomic study was conducted by Nozawa et al. (3). It will be interesting to compare those data with the ones obtained in the present work.

In summary, I believe this is an excellent work. Most of my comments are relatively easy to address. The only one that requires some effort is the comparative proteomic analyses. It would be interesting to know if the differential phosphorylation is due to a direct kinase cascade initiated by dTSSK or, if the reduced phosphorylation is due to lack of synthesis of the respective protein. Besides this point, I would strongly suggest citing mammalian knock-out studies on other TSSKs.

REFERENCES

- 1. P. Shang et al., Functional transformation of the chromatoid body in mouse spermatids requires testis-specific serine/threonine kinases. *J Cell Sci* 123, 331-339 (2010).
- 2. S. Nayyab et al., TSSK3, a novel target for male contraception, is required for spermiogenesis. *Mol Reprod Dev* 88, 718-730 (2021).
- 3. K. Nozawa et al., Testis-specific serine kinase 3 is required for sperm morphogenesis and male fertility. *Andrology*, (2022).
- 4. N. A. Spiridonov et al., Identification and characterization of SSTK, a serine/threonine protein kinase essential for male fertility. *Molecular and cellular biology* 25, 4250-4261 (2005).
- 5. X. Wang et al., Tssk4 is essential for maintaining the structural integrity of sperm flagellum. *Mol Hum Reprod* 21, 136-145 (2015).
- 6. K. N. Jha, S. K. Tripurani, G. R. Johnson, TSSK6 is required for gammaH2AX formation and the histone-to-protamine transition during spermiogenesis. *J Cell Sci* 130, 1835-1844 (2017).
- 7. A. M. Salicioni et al., Testis-specific serine kinase protein family in male fertility and as targets for non-hormonal male contraception. *Biol Reprod* 103, 264-274 (2020).

Reviewer #2 (Remarks to the Author):

In this manuscript, Zhang et. al. report the identification of a TSSK kinase in *Drosophila* (dTSSK) that is crucial for spermiogenesis and male fertility. The authors demonstrated that the kinase activity of

dTSSK is important for histone-to-protamine transition, nuclear shaping, DNA condensation and flagellar organization in spermatids. Using quantitative phosphoproteomics based on TMT-labeling, they find over 828 phosphosites depleted on *Tssk*^{-/-} flies. The depleted phosphosites grouped into several processes required during spermiogenesis such as flagellar organization, motility, spermatid differentiation, etc. *Mdt77F* and *Mst33A*, a protamine-like and a transition protein, respectively, were validated as dTSSK substrates and, importantly, mutation of phosphorylation sites in these proteins were found to impair sperm chromatin condensation and spermiogenesis. Overall, this study presents a range of genetic and biochemical data supporting that dTSSK is a key regulator of spermiogenesis and uncovers important new substrates for this kinase.

The manuscript is well written and organized. The experiments were rigorously planned and executed, and the major claims are supported by abundant and convincing data. The findings elucidate fundamental aspects of the regulation of spermatogenesis, spermiogenesis and male fertility, and will have an important impact in the field of reproductive biology. To improve clarity and robustness of the presented data, it is suggested that the authors provide quantification and respective statistics of several of the immunofluorescence experiments (for example, Figures 3A-C, Figure 4C, and other relevant panels throughout the paper and supplementary figures) to support the claimed changes in protein abundance at the specific stages of spermiogenesis.

Minor issues

1. Typo in line 57: "transinducing".
2. In line 154: "Consistently, flagella were obviously disorganized in spermatid...", please add corresponding graph and statistics.
3. In line 209-210 "...demonstrating that the N-terminal domain of dTSSK Δ N affects protein localization to sperm nuclei." Please, verify if the N-terminal domain of dTSSK contain a consensus NLS and add the analysis to figure S5C.
4. In line 213: "Compared with dTSSK Δ N, the phenotype of dTSSK Δ C-rescued flies was better..." is vague, please add quantification/graphs with proportion to precisely define how these conditions compare.
5. Figure 6B is missing a statistical test (p-value for the differences).
6. In figure 6D, it is necessary to show a western blot for total *Mst77F* to show that the difference is only in p*Mst77F* and not abundance.
7. There are two typos in the word "phospho" in lines 372 and 373, is written phosphor.

Point-by-point response:

REVIEWER COMMENTS

Reviewer #1 (Remarks to the Author):

This work by Zhang et al. explores the functional role of a member of the testis-specific serine kinase family known as TSSKs in flies. These kinases, originally identified in mice have been shown bioinformatically to be present in many other organisms including *C. elegans* and *D. melanogaster*. In this manuscript, the authors focusing in the investigation of the only TSSK found in flies which was named dTSSK. Overall, this is an outstanding study and the first one to use advanced genetic tools to investigate the role of dTSSK, in particular, and TSSKs overall in spermiogenesis. As found with the *Tssk1/Tssk2* double KO, and the single TSSK3 and TSSK6 models, flies lacking dTSSK are sterile. However, this work is able to continue functional studies by conducting rescue experiments with wild type dTSSKs as well with different other genes, including all the human orthologues and several mutants. Overall, this is an outstanding work that transcends the *Drosophila* field and can be used to understand what TSSKs are doing in other systems. More generally, it also offers tools to investigate one of the less known aspects of spermatogenesis which is spermiogenesis.

Despite these positive comments, I have found several aspects that need revision. First, it is surprising the lack of comments on papers describing loss-of-function knock out genetic models in the mouse (see relevant references below). Second, Western blots are shown with cut bands. Third, some of the statements are over-conclusions that deserved to be toned down. Fourth, although the authors have conducted a very good phospho proteomic analysis, addition of a comparative standard proteomic analysis would enhance the impact of this work. Finally, it is important throughout the paper to recognize that decrease in phosphorylation sites does not mean direct phosphorylation by dTSSK. The authors should consider the involvement of kinase cascades in more complex signaling pathways.

Specific Comments.

- The authors have cited most works on the field. It is therefore surprising that they did not cite those showing genetic loss-of-function mice models indicating the essential nature of TSSK1, 2 (1)(double KO), 3 (2, 3) and 6 (4) as well as subfertility in the case of *Tssk4* KO (5).

Answer:

We appreciate your comment and apologize for the oversight in not citing these papers in our manuscript. We have reviewed these studies, and we agree that they are essential to the current understanding of TSSK gene function and their role in male fertility. We have included them in the introduction section of the manuscript (please refer to Line 69-Line 73).

- Fig. 1 A and Fig. 1 E. Complete blots should be shown including MW.

Answer:

Thank you for your feedback and suggestion. We have attached the complete blots for both Fig. 1A and Fig. 1E in the updated supplementary file, including MW markers (please refer to Fig. S1F and S1I). However, we apologize for the error found in Fig. 1A, which was due to the use of dTSSK transgenic flies tagged with Flag-GFP and driven by its own endogenous promoter. This mistake has been corrected in the manuscript.

Figure legend: (A) Raw images of western blot in Figure 1A showing the specific expression of dTSSK protein in testicular tissue. Different tissues (including embryo, head, body, ovary, and testis) were dissected from dTSSK transgenic flies tagged with Flag-GFP and driven by its endogenous promoter. Tissue homogenates were used for WB analysis against anti-Flag, and the observed band size of Flag-GFP-dTSSK is consistent with its predicted size. H3 was used as a loading control. (B) Immunoprecipitation (IP) experiment further validated the specificity of the band and predicted its size in Figure 1A.

- The rescue control with wild type dTSSK is an excellent approach and prepares the reader for the rest of the work.

Answer:

Thank you for acknowledging our approach of using a rescue control with wild type dTSSK and for recognizing its significance in establishing the validity and specificity of our experimental system. We appreciate your positive feedback and agree that this approach serves as a useful comparison for the subsequent experiments conducted in our work.

- Line 184. Although in humans there are only five TSSK genes, in other species such as the mouse, Tssk5 is also present. So, instead of saying the five human Tssk genes were used for rescue, mention that these genes are 1, 2, 3, 4 and 6. (MINOR).

Answer:

Thank you for your comment and for bringing this to our attention. We have revised the relevant section of the manuscript to make this more clear (please refer to Line 190-Line 191).

- Line 189. Analyses of nuclear morphology and histone fate was done with TSSK1 B and TSSK6-rescued flies but not with the others. The authors should make a comment about why TSSK2-, TSSK3- and TSSK4- rescued flies were not further analyzed.

Answer:

Thank you for your feedback. We apologize for any confusion caused by our initial description. Upon further review, we would like to clarify that we actually performed the analyses of nuclear morphology with TSSK1B, 2, 3, 4, and 6. However, we observed that the complementary effect of TSSK1B and TSSK6 was able to restore the morphology of nuclei, while the other three TSSKs could not (Figure S4B). Therefore, we chose to focus on the fate of histone proteins on sperm DNA in *Drosophila* supplemented with TSSK1B as it provided the most informative data for our study. We have revised the manuscript to better describe our results (please refer to Line 194- Line 196).

- Line 207. I agree with the authors that the experiments described here are good. However, I believe that description of this particular experiment should use “strongly suggest” instead of “:demonstrate”. (MINOR)

Answer:

Thank you for your feedback on our manuscript. We appreciate your input and agree that using the phrase "suggest" instead of "demonstrate" would be more appropriate to describe the findings of this particular experiment. We have revised the whole manuscript accordingly.

- Line 216. This sentence is not accurate. I agree with the conclusion but not with the premise. From this reviewer point of view what suggest that the N- and C- terminal-rescued flies are essential for sperm maturation is the fact that these flies can progress to a later spermiogenesis stage upon expression of dTSSK Δ N, and dTSSK Δ C.

Answer:

Thank you for your comment. After carefully reviewing the sentence in question (Line 216), we fully agree with your perspective that it could be rephrased to more accurately reflect the findings of our study. Our results indicate that both the N- and C-terminal domains of dTSSK play essential roles in sperm maturation and male fertility. Accordingly, we have revised the sentence to better convey this key point (please refer to Line 221-Line 223). Thank you again for bringing this to our attention.

- Although difficult to conduct, instead of a gain-of-function rescue, a loss-of-function of dTSSK Δ N, and dTSSK Δ C will be more conclusive regarding the relevance of these fragments.

Answer:

Thank you for your thoughtful review and your suggestion for further investigation. We agree that conducting a loss-of-function experiment with

dTSSK Δ N and dTSSK Δ C would provide valuable information regarding the relevance of these fragments in sperm maturation and male fertility. We appreciate your recognition that such experiments can be challenging to conduct, but we will carefully consider your recommendation as we plan future studies.

- Line 267 to 268. A sentence stating if these identified proteins have mammalian homologues will be helpful for those readers that do not know *Drosophila* nomenclature. (MINOR)

Answer:

Thank you for bringing up the concern about the clarity of our manuscript. We have revised the manuscript to include a sentence indicating whether these proteins have mammalian homologues (please refer to Line 276-Line 277).

- Line 277. Please change “To demonstrate”. Instead use “To evaluate or To test”.

Answer:

Thank you for pointing this out. We corrected it in the manuscript (please refer to Line 287).

- Line 280. It is important that the authors state here (and for the general phosphoproteomic data) that the lack of phosphorylation of a particular protein does not imply that this protein is a dTSSK substrate. It only indicates that dTSSK is part of the pathway. The MAPK pathway is an excellent example to visualize the problem with this conclusion.

Answer:

Thank you for your feedback and suggestion. We agree that it is crucial to clarify that the absence of phosphorylation of a particular protein does not necessarily indicate that it is a direct substrate of dTSSK, but rather that dTSSK is likely part of the pathway regulating its phosphorylation. Other factors may be involved in the regulation of protein phosphorylation, and further studies are needed to fully understand the role of dTSSK in this process. Therefore, we have made corresponding adjustments in the conclusion section of the manuscript (please refer to Line 292/ Line 298/ Line 301/Line 311).

- Line 283. The Western blot reveals absence of the phosphorylation site. However, in the absence of a control blot with an antibody that recognize total Mst77F, the authors cannot claim that the lack of signal is due to lack of phosphorylation or to reduced Mst77F protein expression. Ideally, this control should be added. In Fig. 6 E, this point is addressed by heterologous expression of a tagged Mst77F. So, overall, this comment is MINOR.

Answer:

Thank you for your feedback on the Western blot analysis. Unfortunately, we were unable to present the total amount of Mst77 protein in Figure 6D due to the unavailability of high-quality Mst77 antibodies. Although we did not include

a control blot for total Mst77F protein, we addressed this concern by performing heterologous expression of tagged Mst77F in Fig. 6E, which provides additional evidence supporting our conclusion.

- Ideally, a proteomic analysis should be added to complement the results. It is not clear if the phospho peptides observed are due to decreased phosphorylation or decreased protein expression. Addition of a standard comparative proteomic analysis would be informative of the role of dTSSK in controlling protein synthesis and will increase the impact of this work.

Answer:

Thank you for your insightful feedback on our results. We agree that a comparative proteomic analysis would be informative and help distinguish between decreased phosphorylation and decreased protein expression. To address this concern, we conducted a proteomic analysis in both wild-type and dTSSK mutant flies, comparing the protein expression differences between their testes. We have included these data in the revised manuscript and provided all proteomic data in the supplementary file (please refer to Line 263-Line 266 and Supplementary Data 1).

For example, our results show that the down-regulated phosphorylated proteins, Mst77F and Mst33A, detected by phosphoproteomic profiling, did not exhibit decreased protein expression in the dTSSK mutants according to our proteomic analysis. We appreciate your feedback and believe that these additional findings will significantly enhance the impact of our study.

- Line 308. This conclusion is not accurate. Don't take me wrong, these experiments are excellent; however, they do not "demonstrate" direct phosphorylation. The co-expression experiment confirms that when Mst77F is present dTSSK mediates its phosphorylation through an unknown pathway. It is silent regarding direct phosphorylation because other kinases are present. Demonstration of direct phosphorylation would require an in vitro assay with purified protein. However, even this experiment would be not completely conclusive because it will not be possible to state that direct phosphorylation occurs in vivo. In line 348, a more direct experiment is described. The authors, here, immunoprecipitated the endogenous dTSSK from cell-free extracts and conduct an in vitro kinase assay. Although this experiment limits the amount of proteins associated to dTSSK, still cannot discard that dTSSK IP brings down other associated kinases responsible for this phosphorylation.

Answer:

Thank you for your valuable feedback. We acknowledge that the co-expression experiment presented in Fig. 5B confirms that dTSSK is involved in the phosphorylation of Mst77F, but does not necessarily imply direct phosphorylation, as other kinases may also be present.

As you have rightly pointed out, the in vitro kinase assay with endogenous dTSSK IP provides a more direct approach to address the issue of direct phosphorylation. However, we also acknowledge your comment regarding the possibility of other associated kinases being responsible for the observed phosphorylation. We have revised the description of this conclusion in the

revised manuscript (please refer to Line 311 and Line 319), taking into consideration your valuable feedback.

- Line 344. It is important that the authors try to describe mammalian orthologues (if available).

Answer:

Thank you for your helpful feedback on this point. We agree that it is important to consider mammalian orthologues of *Drosophila* Mst33A. We have made an effort to identify mammalian homologues, but unfortunately, we could not find a clear match using sequence blast searches. However, we suggest that Mst33A may share a similar function to the mammalian transition proteins TP1/TP2 based on their similar localization and biological function (Grimes et al. 1977 PMID: 923663, Marvin et al., 2003 PMID: 12743712; Want et al., 2019 PMID: 31649732). We acknowledge that further studies are needed to confirm this hypothesis and identify potential mammalian homologues of Mst33A. We have included this discussion in the revised manuscript. Thank you again for your insightful comments and suggestions.

- Line 365. Please change the word “determine” to “evaluate/test/investigate” (or another word that does not imply that the authors expect their hypothesis to be correct).

Answer:

Thank you for your helpful suggestion. We have incorporated the change you recommended, specifically replacing the word "determine" with "evaluate" in the revised manuscript (please refer to Line 377).

- Fig.8 C and D. Add whole Western blots in a supplementary figure.

Answer:

Thank you for your valuable feedback on Fig.8 C and D. We appreciate your suggestion to include the whole Western blot images in a supplementary figure. We would like to clarify that Fig. 8D is a fertility statistic and not a Western blot image, and apologize for any confusion in our previous manuscript. Here, we speculate that you may be referring to Fig. 8B and Fig. 8C, rather than Fig. 8C and Fig. 8D.

In the revised manuscript, we have included a supplementary figure that shows the entire Western blot images for Fig. 8B and Fig. 8C, with each lane clearly labeled and a molecular weight marker included for reference (please refer to Fig. S11A and Fig. S11B).

For Fig. S11A: Raw image of Western blot analysis in Fig. 8B.

For Fig.S11B: The figures below showing the raw images of WB blots from two repeated experiments (Repeat 1 is the raw image of WB analysis in Fig. 8C).

Figure legend: Raw image of WB analysis showing phosphorylation of dTSSK in testicular extracts of dTSSK-Flag, dTSSK^{S17A}-Flag, dTSSK^{S18A}-Flag, and dTSSK^{S17A&S18A}-Flag flies. Ppase treatment removes the phosphorylation modification of dTSSK-Flag and dTSSK^{S17A}-Flag.

- Line 376. Line 365. Please change the word “demonstrate” to “evaluate/test/investigate”. (or another word that does not imply an expected result).

Answer:

Thank you for your comment. We have carefully considered your suggestion and made the change you suggested, changing the word "demonstrate" to "investigate" in the revised manuscript (please refer to Line 377 and Line 388).

- Line 403. This expression is correct. The authors should use it throughout the paper. The word "potential" here is essential.

Answer:

Thank you for your comment. We appreciate your feedback on the use of the term "potential" in our manuscript. Upon careful consideration of your suggestion, we have revised the manuscript and used the term "potential" consistently throughout the paper (please refer to Line 476/ Line 490/ Line 494/ Line 496/Line 499).

- Line 411. Citation needed for the mammalian case.

Answer:

Thank you for your comment. We appreciate your feedback and have added relevant references to support the mammalian case in the revised manuscript (please refer to Line 423).

- Line 415: Not clear why the authors said that dTSSK function is distinct from the functional role of TSSKs in mammals. First, there are 6 TSSKs in the mouse and five in humans, their role in spermiogenesis is not well defined, so, it is difficult to state that they don't have similar functions. A plausible hypothesis is that the dTSSK role divided in mammals require more than one TSSK. Second, although rescue is not complete, results in the present work showed that, at least TSSK1B and TSSK6 are able to rescue many defects observed in the dTSSK KO flies. Third, it is not possible to discard that expression of a combination of mammalian TSSKs would result in complete rescue of the fertility phenotype in flies.

Answer:

We greatly appreciate your valuable feedback and have taken it into consideration. We have revised the sentence to reflect your suggestions (please refer to Line 427).

- Line 425. This sentence "In vitro experiments consistently fail..." is not clear. Which in vitro experiments, the authors are referring to? Are they referring to the IP of dTSSKs from flies? Please explain in the text.

Answer:

Thank you for your question and feedback. We apologize for any confusion caused by our original sentence. To clarify, we were specifically referring to "the previous studies on the function of TSSKs", which do not provide clear evidence of a link between kinase activity and male sterility. We have revised the sentence to make this point clearer (please refer to Line 437).

- Line 446. It is very important here to cite Jha et al. JBC paper showing that in the absence of TSSK6, there are problems with histone replacement, reminiscent of the findings in the present work (6).

Answer:

Thank you for your suggestion. In the absence of TSSK6, Jha et al. observed

difficulties in histone replacement on sperm DNA, which aligns with our findings. Thus, we have included a citation to Jha et al.'s paper in our manuscript to acknowledge the contribution of their research and to provide further context for our study (please refer to Line 457-Line 458).

- Line 455. In the TSSK6 KO model, H2AX (a phosphorylated form of H2) is not found during spermiogenesis. However, in vitro experiments show that there is no direct phosphorylation suggesting that this phosphorylation is part of a more complex kinase cascade.
- The hypothesis of phosphorylation of ubiquitin-modifying enzymes is attractive.

Answer:

Thank you for your comment. It is interesting to note that in the TSSK6 KO model, H2AX is not found during spermiogenesis, despite the lack of direct phosphorylation in vitro. The hypothesis of phosphorylation of ubiquitin-modifying enzymes is indeed an attractive possibility that warrants further investigation. It is possible that TSSKs may play a role in regulating the activity of these enzymes, which in turn may affect the histone replacement during spermiogenesis. Further studies are needed to confirm this hypothesis and to elucidate the underlying molecular mechanisms.

- As a final comment, many authors have considered TSSKs as targets for contraception (7). A couple of sentences mentioning this point will be of interest for many readers.

Answer:

Thank you for bringing up this important point. We agree that the potential use of TSSKs as targets for contraception is an interesting and relevant topic. We have included a brief discussion of this in the revised manuscript for the benefit of our readers (please refer to Line 523- Line 528).

For instance:

"...TSSKs have been suggested as promising targets for male contraception (20). Our current study provides further mechanistic evidence that TSSKs play a critical role in sperm development and maturation, and their inhibition could disrupt spermiogenesis and impede the production of functionally competent sperm. It is worth noting that although promising progress has been made in developing contraceptive drugs using TSSK inhibitors in animals, further research is needed to establish their efficacy and safety in humans."

- A previous phosphoproteomic study was conducted by Nozawa et al. (3). It will be interesting to compare those data with the ones obtained in the present work.

Answer:

Thank you for your valuable comment. We appreciate your suggestion to compare our phosphoproteomic data with the study conducted by Nozawa et al. In fact, we have observed that some homologous proteins regulated by TSSK3 in mice also showed down-regulation in the phosphorylation level in

dTSSK mutant flies, such as *kl-3* (*Dnah8* homolog in mice), *Osbp2* (*Osbp* homolog in mice), and *TLL4B* (*Tll5* homolog in mice). In response to your suggestion, we have included a brief discussion in the revised manuscript that highlights the similarities between our results and those obtained by Nozawa et al. We hope that our revised manuscript now provides a more comprehensive and insightful analysis of our findings (please refer to Line 503-Line 510).

In summary, I believe this is an excellent work. Most of my comments are relatively easy to address. The only one that requires some effort is the comparative proteomic analyses. It would be interesting to know if the differential phosphorylation is due to a direct kinase cascade initiated by dTSSK or, if the reduced phosphorylation is due to lack of synthesis of the respective protein. Besides this point, I would strongly suggest citing mammalian knock-out studies on other TSSKs.

Answer:

Thank you for your positive feedback and constructive comments on our work. We are glad to hear that you found our work excellent. Regarding the comparative proteomic analysis, we appreciate your suggestion and have performed further proteomics experiments in wild-type and dTSSK mutant flies to address this question (please refer to Line 263-Line 266 and Supplementary Data 1). Moreover, we have included relevant references on mammalian knock-out studies on other TSSKs in the revised manuscript, as suggested by you (please refer to Line 69-Line 73).

We appreciate your valuable feedback and thank you once again for taking the time to review our manuscript.

REFERENCES

- 1. P. Shang et al., Functional transformation of the chromatoid body in mouse spermatids requires testis-specific serine/threonine kinases. *J Cell Sci* 123, 331-339 (2010).
- 2. S. Nayyab et al., TSSK3, a novel target for male contraception, is required for spermiogenesis. *Mol Reprod Dev* 88, 718-730 (2021).
- 3. K. Nozawa et al., Testis-specific serine kinase 3 is required for sperm morphogenesis and male fertility. *Andrology*, (2022).
- 4. N. A. Spiridonov et al., Identification and characterization of SSTK, a serine/threonine protein kinase essential for male fertility. *Molecular and cellular biology* 25, 4250-4261 (2005).
- 5. X. Wang et al., Tssk4 is essential for maintaining the structural integrity of sperm flagellum. *Mol Hum Reprod* 21, 136-145 (2015).
- 6. K. N. Jha, S. K. Tripurani, G. R. Johnson, TSSK6 is required for gammaH2AX formation and the histone-to-protamine transition during spermiogenesis. *J Cell Sci* 130, 1835-1844 (2017).
- 7. A. M. Salicioni et al., Testis-specific serine kinase protein family in male fertility and as targets for non-hormonal male contraception. *Biol Reprod* 103, 264-274 (2020).

Reviewer #2 (Remarks to the Author):

In this manuscript, Zhang et. al. report the identification of a TSSK kinase in *Drosophila* (dTSSK) that is crucial for spermiogenesis and male fertility. The authors demonstrated that the kinase activity of dTSSK is important for histone-to-protamine transition, nuclear shaping, DNA condensation and flagellar organization in spermatids. Using quantitative phosphoproteomics based on TMT-labeling, they find over 828 phosphosites depleted on Tssk^{-/-} flies. The depleted phosphosites grouped into several processes required during spermiogenesis such as flagellar organization, motility, spermatid differentiation, etc. Mdt77F and Mst33A, a protamine-like and a transition protein, respectively, were validated as dTSSK substrates and, importantly, mutation of phosphorylation sites in these proteins were found to impair sperm chromatin condensation and spermiogenesis. Overall, this study presents a range of genetic and biochemical data supporting that dTSSK is a key regulator of spermiogenesis and uncovers important new substrates for this kinase.

The manuscript is well written and organized. The experiments were rigorously planned and executed, and the major claims are supported by abundant and convincing data. The findings elucidate fundamental aspects of the regulation of spermatogenesis, spermiogenesis and male fertility, and will have an important impact in the field of reproductive biology. To improve clarity and robustness of the presented data, it is suggested that the authors provide quantification and respective statistics of several of the immunofluorescence experiments (for example, Figures 3A-C, Figure 4C, and other relevant panels throughout the paper and supplementary figures) to support the claimed changes in protein abundance at the specific stages of spermiogenesis.

Answer:

Thank you for your thoughtful comments and positive feedback on our manuscript. We appreciate your suggestion regarding the presentation of the data and have carefully addressed this issue in our revised manuscript. Specifically, we have included quantification and statistical analysis of the immunofluorescence experiments in relevant figures, including Figures 3A-C, Figure 4C, and other relevant panels throughout the paper and supplementary figures (please refer to Fig. S3A-S3C, S4C, S5C, S6F and S9C). We believe that these additions will enhance the clarity and robustness of our results.

Minor issues

1. Typo in line 57: “transinducing”.

Answer:

Thank you for bringing this to our attention. We apologize for the error and we have corrected it to "...by inducing H4 acetylation in trans" in the revised version of the manuscript.

2. In line 154: “Consistently, flagella were obviously disorganized in spermatid...”, please add corresponding graph and statistics.

Answer:

Thank you for your feedback and suggestion. We agree that adding corresponding graphs and statistics would increase the robustness of our data. We have analyzed the data and provide the requested quantification and statistics for the flagella disorganization in spermatids (please refer to Fig. S2B).

3. In line 209-210 “...demonstrating that the N-terminal domain of dTSSK Δ N affects protein localization to sperm nuclei.” Please, verify if the N-terminal domain of dTSSK contain a consensus NLS and add the analysis to figure S5C.

Answer:

Thank you for your valuable suggestion. We have carefully examined the N-terminal domain of dTSSK and found that it does not contain a consensus nuclear localization signal (NLS) sequences, based on the results of several online programs (<https://roslab.org/services/nlsdb/>; <https://www.novoprolabs.com/tools/nls-signal-prediction>).

4. In line 213: “Compared with dTSSK Δ N, the phenotype of dTSSK Δ C-rescued flies was better...” is vague, please add quantification/graphs with proportion to precisely define how these conditions compare.

Answer:

Thank you for your valuable feedback. We appreciate your suggestion to add quantification/graphs to precisely define the comparison between dTSSK Δ N and dTSSK Δ C-rescued flies at late canoe stage of spermiogenesis. We have included these statistical data in the revised manuscript (please refer to Fig. S5C).

5. Figure 6B is missing a statistical test (p-value for the differences).

Answer:

Thank you for your comment. We have now conducted the statistical test and added the corresponding p-value to the figure (please refer to Fig. 6B).

6. In figure 6D, it is necessary to show a western blot for total Mst77F to show that the difference is only in pMst77F and not abundance.

Answer:

Thank you for your feedback on our Western blot analysis. Unfortunately, we were unable to present the total amount of Mst77 protein in Figure 6D due to the unavailability of high-quality Mst77 antibodies. However, we have addressed this concern by performing a heterologous expression of tagged

Mst77F in Fig. 6E. This experiment provides further evidence to support our conclusion that the absence of phosphorylation site in Mst77F is due to the absence of dTSSK protein.

7. There are two typos in the word “phospho” in lines 372 and 373, is written phosphor.

Answer:

Thank you for bringing this to our attention. We apologize for the typographical error in the manuscript. We have carefully reviewed the text and have corrected the errors read "phospho" instead of "phosphor" (please refer to Line 384-Line 385). Thank you for your attention to detail and for helping us improve the quality of our manuscript.

** See Nature Portfolio's author and referees' website at www.nature.com/authors for information about policies, services and author benefits.

This email has been sent through the Springer Nature Tracking System NY-610A-NPG&MTS

REVIEWERS' COMMENTS

Reviewer #1 (Remarks to the Author):

I appreciate the authors willingness to address every point of the review. As mentioned for the previous version, this is an outstanding contribution to our understanding of TSSK role in reproduction.

Reviewer #2 (Remarks to the Author):

The authors have addressed my concerns.

Point-by-point response:

REVIEWER COMMENTS

Reviewer #1 (Remarks to the Author):

I appreciate the authors willingness to address every point of the review. As mentioned for the previous version, this is an outstanding contribution to our understanding of TSSK role in reproduction.

Reply:

Thank you for taking the time to review our manuscript and for your positive feedback on our revised version. We are glad to hear that you appreciate our efforts to address each point mentioned in the review.

Furthermore, we are also appreciate your acknowledgement that our study represents an outstanding contribution to our understanding of the role of TSSK in reproduction. We hope that our research can contribute to further advancements in the field.

Thank you again for your valuable feedback and for your time in evaluating our work.

Reviewer #2 (Remarks to the Author):

The authors have addressed my concerns.

Reply:

Thank you for taking the time to review our manuscript. We are pleased to hear that we were able to address your concerns. We appreciate your feedback and suggestions, which helped us to improve the quality of our work.